# A continued role of Short-Lived Climate Forcers under the Shared Socioeconomic Pathways

*Marianne T. Lund[1*], Borgar Aamaas[1], Camilla W. Stjern[1], Zbigniew Klimont[2], Terje K. Berntsen[1,3], Bjørn H. Samset[1]*

*1 CICERO, Center for International Climate Research, Oslo, Norway*

*2 International Institute for Applied Systems Analysis (IIASA), Laxenburg, Austria*

*3 Department of Geosciences, University of Oslo, Oslo, Norway*

*\*Corresponding author: m.t.lund@cicero.oslo.no*

## Abstract

Mitigation of non-$CO_2$ emissions plays a key role in meeting the Paris Agreement ambitions and Sustainable Development Goals. Implementation of respective policies addressing these targets mainly occur at sectoral and regional levels and designing efficient mitigation strategies therefore relies on detailed knowledge about the mix of emissions from individual sources and their subsequent climate impact. Here we present a comprehensive dataset of near- and long-term global temperature responses to emissions of $CO_2$ and individual short-lived climate forcers (SLCFs) from 7 sectors and 13 regions - for present-day emissions and their continued evolution as projected under the Shared Socioeconomic Pathways. We demonstrate the key role of $CO_2$ in driving both near- and long-term warming, and highlight the importance of mitigating methane emissions, from agriculture, waste management and energy productions, as the primary strategy to further limit near-term warming. Due to high current emissions of cooling SLCFs, policies targeting end-of-pipe energy sector emissions may result in net added warming unless accompanied by simultaneous methane and/or $CO_2$ reductions. We find that SLCFs are projected to play a continued role in many regions, particularly those including low- to medium-income countries, under most of the SSPs considered here. East Asia, North America and Europe remain the largest contributors to total net warming until 2100, regardless of scenario, while South Asia and Africa south of the Sahara overtake Europe by the end of the century in SSP3-7.0 and SSP5-8.5. Our dataset is made available in an accessible format, aiming also at decision-makers, to support further assessments of the implications of policy implementation at the sectoral and regional scales.

## 1 Introduction

At the core of any strategy for sustained, long-term abatement of climate change are strong reductions in emissions of $CO_2$ and other long-lived greenhouse gases (LLGHGs). However, most anthropogenic activities emit a suite of additional species, with a range of climate impacts, commonly termed short-lived climate forcers (SLCFs). While differing in characteristics and contribution to temperature change, their common feature of a much shorter atmospheric residence time compared to LLGHGs has resulted in significant discussion of the role of SLCF mitigation in strategies to reduce climate change, in particular to limit near-term warming (e.g.,Bowerman et al., 2013; Pierrehumbert, 2014; Rogelj et al., 2015; Shindell et al., 2012; Shoemaker et al., 2013; Stohl et al., 2015).

Many assessments have placed particular emphasis on the subset of SLCFs with a warming impact on climate, namely black carbon (BC), methane ($CH_4$) and tropospheric ozone (sometimes collectively referred to as short-lived climate pollutants, or SLCPs) (e.g., AMAP, 2015; CCAC, 2019; UNEP, 2017). Assuming effective abatement of SLCPs, some studies estimate a reduction in global temperature increase of 0.2-0.5°C by mid-century (e.g., Shindell et al., 2012). More recent work suggest that some of these early estimates may overestimate the effect of SLCP mitigation (Rogelj et al., 2014; Smith & Mizrahi, 2013; Stohl et al., 2015; Takemura & Suzuki, 2019). While results from early studies brought some concern that the attractiveness of SLCP mitigation could lead to delayed action on $CO_2$ emissions, most scientific studies emphasize that SLCP measures should only be considered complementary to early and stringent $CO_2$ mitigation for the achievement of long-term climate goals (Ramanathan & Carmichael, 2008; Rogelj et al., 2014).

SLCF mitigation may also give rise to potential trade-offs. Due to co-emission, any given mitigation measure or policy can affect a broad range of species. The combinations may, however, vary significantly between sources and mitigation strategies motivated by, and designed to address, different societal challenges. For instance, many SLCFs are tightly linked to air quality (Anenberg et al., 2012; Lelieveld et al., 2015; Shindell et al., 2012) and sustainable development (Haines et al., 2017; UNEP, 2019), in addition to their climate impacts. The numerous environmental and societal co-benefits of SLCF reductions are well recognized but may lead to adverse climatic consequences (Arneth et al., 2009). While some SLCFs with a warming contribution to temperature change can, in part, be mitigated individually (in particular methane), improving air quality requires consideration of all relevant species. Removal of all present-day anthropogenic aerosols may add as much as 0.5°C of additional global near-term warming according to recent work (Hienola et al., 2018; Samset et al., 2018; Aamaas et al., 2019). Due to co-emission, species such as sulfur dioxide ($SO_2$) are also commonly affected by measures to reduce climate warming even if these have LLGHGs as the primary target. Hence, while it remains clear that deep reductions in emissions of methane and BC play a key role in pathways for global emissions that limit global warming to 1.5°C and 2°C warming (Harmsen et al., 2019; Rogelj et al., 2015; Rogelj et al., 2018; Shindell & Smith, 2019; Xu & Ramanathan, 2017), co-emitted species such as sulfate need to be carefully considered.

A key characteristic of SLCFs is that the composition of emissions, as well as their subsequent radiative forcing, can vary significantly between individual emission sources (Bond et al., 2013;

Lund et al., 2014b; Persad & Caldeira, 2018; Unger et al., 2010). While previous scenarios for long-term evolution of aerosols and ozone precursor emissions projected a general, rapid decline even in pathways with high climate forcing and GHG levels (Gidden et al., 2019; Rao et al., 2017), the most recent generation scenarios, the Shared Socioeconomic Pathways (SSPs) (O'Neill et al., 2014; Riahi et al., 2017), exhibit a much larger spatiotemporal heterogeneity in projections of these emissions. Additionally, the SSPs provide a framework for combining future climate scenarios (Representative Concentration Pathways – RCPs) with socioeconomic development, and hence more detailed information about plausible future evolutions of society and natural systems. Up-to-date and detailed knowledge of the climate impact of individual emission sources is critical for the design of effective mitigation strategies and to provide decision makers with more integrated guidance on how to best address linkages between climate, sustainable development and air quality in policy processes (Melamed et al., 2016). While studies comparing and quantifying the impacts of SLCFs and $CO_2$ exist, they differ in selection of sectors and/or regions, methodology and emission inventory, making direct comparison difficult (e.g., Harmsen et al., 2019; Kupiainen et al., 2019; Lund et al., 2014a; Sand et al., 2015; Unger et al., 2010). Furthermore, studies often consider only the equilibrium effect of present-day emissions, emission pulses or very simplified scenarios.

In the present work, we provide a comprehensive and updated investigation of the contribution to near- and long-term global temperature impacts from individual SLCF and LLGHG emissions. We first quantify the temperature response to an idealized pulse of present-day emissions to demonstrate the methodology and temporal behavior of the various emitted species, focusing on both added benefits and trade-offs offered by SLCF mitigation. Then we calculate the future evolutions of temperature impacts as they are projected to develop under the pathways for future socioeconomic development, climate policy and air pollution described by the SSP-RCP scenarios. The temperature impact is calculated for seven economic sectors and 13 source regions, accounting for best available knowledge and geographical dependence of the forcing efficacy of different SLCFs, thereby providing a more detailed breakdown than previous literature. By making our full data set openly available, we aim to provide a toolkit for further studies of the implications of policy implementation at the sectoral and regional level, demonstrating the potential for such applications for a set of idealized sectoral emission reduction packages.

## 2 Methodology

Using the concept of Absolute Global Temperature change Potential (AGTP) (Shine et al., 2005), we calculate the global-mean temperature response over time to emissions of $CO_2$, $CH_4$, ammonia ($NH_3$), BC, OC, $SO_2$, the ozone precursors nitrogen oxide (NOx), carbon monoxide (CO) and volatile organic compounds (VOCs) from 7 sectors and 13 regions (Fig. 1).

2.1 Calculations of global and regional AGTPs

The AGTP is an emission metric-based emulator of the climate response, and a well-established method that enables us to quantify and compare global temperature impacts of a large number of sources and scenarios in a transparent and, in terms of computer resources, cost-effective

manner. The approach is described in detail in the literature (Fuglestvedt et al., 2010; Shine et al., 2005; Aamaas et al., 2013); here we give a brief outline.

The ATGP gives the global-mean surface temperature response per kg species emitted as a function of time after an emission pulse, i.e., an instantaneous one-off emission. At time $H$ after the emission, the AGTP for species $i$ is given (for each sector and region) by:

$$AGTP_i(H) = \int_{t=0}^{H} F_i(t)IRF_T(H-t)dt \tag{1}$$

where $F_i$ is the radiative efficiency. Emissions of SLCFs can have both direct and indirect radiative effects. For BC, OC and $SO_2$ we account for the direct, semi-direct and indirect RF as described below. AGTPs for NOx, CO and VOC includes the forcing due to tropospheric ozone production and (for NOx) nitrate aerosol formation, as well as the longer-term effect on methane lifetime and methane-induced ozone loss. The AGTP for methane includes the direct forcing, as well as the effect of OH-induced changes in its lifetime and effects on tropospheric ozone and stratospheric water vapor. See Aamaas et al. (2013) for details and analytical expressions for the AGTP of individual species.

For $CO_2$ and methane, we calculate the global-mean $F$ for year 2014 global concentrations (i.e., the year that is considered present-day in our emissions data – see below) using the equations from Etminan et al. (2016). Compared to the approach used on the IPCC Fifth Assessment report (AR5) (Myhre et al., 2013), this increases the radiative efficiency of methane by 14%. For $NH_3$, we use the IPCC AR5 best estimate for global mean radiative efficiency for all regions. For the remaining short-lived species, we derive values of $F_i$ that depend on the location of the emission and calculate region-specific AGTPs for BC, OC, $SO_2$, and the ozone precursors. The regional radiative efficiencies (i.e., the global radiative forcing per unit of regional emissions) for BC, OC, sulfate, nitrate and ozone (in response to NOx, CO and VOC) are derived using radiative kernels (Samset & Myhre, 2011) and atmospheric concentrations from simulations performed with the global chemistry transport model OsloCTM3 (Søvde et al., 2012) for the second phase of the Hemispheric Transport of Air Pollution (HTAP2) (Janssens-Maenhout et al., 2015). Details about the chemistry and aerosol parameterizations and properties can be found in Lund et al. (2018). In addition to their direct radiative effects, aerosols also affect the energy balance through modifications of clouds and atmospheric heating rates (indirect and semi-direct effects). To account for the additional negative RF resulting from aerosol-cloud interactions, we scale the AGTP of $SO_2$ by a factor of 2.1 based on the ratio of total global RF of sulfate to that due to direct effects alone from the IPCC AR5 (Myhre et al., 2013). Due to lack of available information about geographical dependence of the radiative efficiency, the same scaling factor is applied for all regions, recognizing that this is a simplification as the indirect effect also likely varies with location of emission. We also account for the semi-direct effect of BC (i.e., the rapid adjustments of the atmosphere to the local heating (Smith et al., 2018)). Here we use the multi-model ratio between semi-direct and direct BC RF from Stjern et al. (2017) and calculate an average adjustment factor for the rapid adjustments of -15%. This is then applied to the AGTP of BC for all regions, except South Africa where Stjern et al. (2017) found a small positive forcing from rapid adjustments. Radiative forcing of BC deposition on snow and ice is not included in our estimates.

$IRF_T$ in Eq.1 is the impulse response function used to estimate the temperature response to a
given radiative forcing:
$$IRF_T(t) = \lambda \sum_{j=1}^{J} \frac{c_j}{d_j} \exp\left(-\frac{t}{d_j}\right)$$ (2)
where $c_j$ and $d_j$ are constants and timescales of the fast and slow model of the climate system
response, respectively, and $\lambda$ is the equilibrium climate sensitivity (ECS). An IRF is also used
to represent the atmospheric decay of $CO_2$. Several different IRFs exist in the literature. Here
we use the $IRF_T$ from Geoffroy et al. (2013) (G13) and the $IRF_{CO2}$ from Joos et al. (2013).
Values of $c_j$, $d_j$ and $\lambda$ derived from the analytical solution of the two-layer energy balance model
used by G13 are given in Table 1. Compared to the $IRF_T$ from Boucher and Reddy (2008)
(B&R08) used in the bulk of previous metrics studies including IPCC AR5, G13 has shorter
timescales and yields a lower ECS (0.885 K $(Wm^{-2})^{-1}$) compared to 1.06 K $(Wm^{-2})^{-1}$) from
B&R08. To place our values in the context of previous literature and explore sensitivities to the
choice of IRFs, we perform additional calculations using different combinations of $IRF_T$ and
$IRF_{CO2}$ – see section Sect. 1 of the Supplementary Information (SI).
Finally, we consistently account for the climate-carbon feedback (CCf) in the AGTPs. The
$IRF_{CO2}$, derived from complex models, implicitly includes the CCf. However, this is not the
case for other components. This inconsistency was first highlighted in Myhre et al. (2013),
where a first attempt to include the CCf was made for halocarbons based on an earlier study by
Collins et al. (2013). This method has since been refined. Here we use the framework developed
by Gasser et al. (2017) where a separate IRF for the CCf was derived using the simple Earth
system model OSCARv2.2. This IRF is used to calculate a $\Delta AGTP_i(H)$ which is then added to
the $AGTP_i(H)$ without CCf. The difference between this method and the approach taken by
Myhre et al. (2013) is discussed in Gasser et al. (2017). We also perform a sensitivity test to
quantify the impact on our estimated temperature responses of excluding the CCf – see Sect.
4.1. Furthermore, as different methods to account for the CCf exist in the literature, we provide
both sets of AGTPs for further use (see "Data Availability").
2.2 Emission data and temperature response calculations
As described above, we investigate the role and global temperature impacts of SLCF and $CO_2$
from two different perspectives. First, the AGTPs at two given time horizons $H$ (here 10 and
100 years) are multiplied by year 2014 emissions from the Community Emission Data System
(CEDS) (Hoesly et al., 2018) for each species, sector and region. The result is the near- and
long-term global temperature response, $\Delta T_i(H)$, to present-day regional and sectoral emissions.
Next, we quantify the temperature response to temporally evolving emissions from 1900 to
2100. The AGTP framework can readily be extended from pulse-based calculations since any
scenario can be viewed as a series of pulse emissions and analyzed through convolution
(Aamaas et al., 2013). The temperature response $\Delta T$ at time t for species i is (for each region
and sector) given by:

$$\Delta T_i(t) = \int_0^t E_i(t') AGTP_i(t - t') dt'$$

Importantly, the AGTPs are linear in that they do not account for the potential changes in radiative efficiency with changing background pollution levels – see Sect. 4 for further discussion.

Historical emissions are from the CEDS database, while future emissions follow the SSP-RCP scenarios. Gridded and harmonized emissions are available via ESFG from the Integrated Assessment Modeling Community (IAMC) for nine SSP-RCP combinations that form the core of the Coupled Model Intercomparison Project Phase 6 (CMIP6) experiments (Gidden et al., 2019): SSP1-1.9, SSP1-2.6, SSP2-4.5, SSP3-7.0, SSP3-7.0 lowNTCF, SSP4-3.4, SSP4-6.0, SSP5-3.4, and SSP5-8.5. The gridded SSP-RCP data product, including the methodology for country and sector level emission mapping, is documented by Feng et al. (2020). We extract regional emission scenarios using the geographical definitions and spatial mask from HTAP2 (Janssens-Maenhout et al., 2015). Furthermore, we consider the energy (ENE), agriculture (AGR), waste (WST), residential (RES), industry plus solvents (IND), transport (TRA) and shipping (SHP) sectors, as they are defined in the CEDS-SSP inventory (Feng et al., 2020; Hoesly et al., 2018). Due to the large spread in historical estimates and lack of emissions consistent with CEDS, we do not include $CO_2$ emissions due to land-use/land cover. Additionally, agricultural waste burning is excluded as these are more difficult to mitigate and estimates of future $CO_2$ emissions are not available.

2.3 Uncertainties

We establish a range in total net global-mean temperature response on 10- and 100-year time scales due to uncertainties in radiative forcing by performing a Monte Carlo analysis. Each RF mechanism is treated as a random variable, following a probability density function (PDF) defined based on existing literature, and the distribution for the total RF is derived by summing the individual PDFs, i.e., assuming that each RF mechanisms is independent. For the aerosols and their precursors, we use the multi-model results from the AeroCom Phase II experiment (Myhre et al., 2013a), while for $CO_2$ $NH_3$, and ozone precursors, we use the uncertainties from the IPCC AR5 (Myhre et al., 2013b). For further details, see Aamaas et al. (2019) and Lund et al. (2017). Our temperature responses are also influenced by uncertainties in emissions and climate sensitivity. A comprehensive analysis of uncertainty in all three factors is challenging due to lack of data, but the potential impact is discussed in Sect. 4.

**3 Results**

**3.1 Near- and long-term temperature response to current emissions**

We first discuss the global mean surface temperature response to one year of present-day (i.e., year 2014) emissions, for global total emissions and broken down by key contributing sectors and geographical source regions as shown in Fig.2. While we here select 10- and 100-year time horizons to represent near- and long-term impacts, we recognize that other choices may affect

the relative importance, and even sign, of the temperature response from some of the SLCFs or
be more relevant for certain applications. For this reason, we also provide the full time series
of our AGTPs (see Data Availability).
Globally, current emissions result in an approximate balance between cooling and warming
SLCFs in the near-term, with main warming contributions from BC and $CH_4$ and cooling from
$SO_2$ and NOx (Fig.2a). The total net effect after 10 years is therefore only slightly larger than
that due to $CO_2$ alone. As the impact of the SLCFs decays over years to decades upon emission,
the total net temperature impact after 100 years is predominantly determined by $CO_2$. As clearly
seen in Fig. 2a, $CO_2$ emissions also cause a notable contribution to near-term warming. While
both of these features are well known in the scientific community, the role of $CO_2$ as driver also
of near- term warming is not always fully acknowledged in the discussions of LLGHGs versus
SLCFs.
Differences in the mix of emissions result in net impacts on global temperature that vary
significantly, in both magnitude and sign, between sectors and regions. Of the economic sectors,
energy (ENE), agriculture (AGR), and waste management (WST) give the largest net near-term
warming (i.e., after 10 years) (Fig. 2b). For AGR and WST, this is a result of strong methane-
induced warming. The energy sector (ENE) is also characterized by a significant warming due
to methane (originating from fossil fuel mining and distribution), as well as $CO_2$, but also by a
considerable cooling from high emissions of $SO_2$. Our results hence reinforce the importance
of methane as a driver of near-term warming but show that the net effect on global temperature
of SLCF mitigation may be small in the case of the energy sector if simultaneous reductions in
$SO_2$ take place. A particular feature of the energy sector, however, is that a significant portion
of methane mitigation from oil and gas (production and distribution) can be done independently
from other energy-related (combustion) emissions. An explicit distinction between production
and combustion emissions was not available in the gridded CEDS inventory, but, as illustrated
in Sect. 3.2, mitigation strategies targeting one category or the other can result in distinctly
different temperature outcomes. Global emissions from industry (IND) and shipping (SHP)
cause a net cooling impact despite a considerable warming from $CO_2$ emissions. In the long
term, the net impact of AGR and WST is small, while energy is the largest individual
contributor to warming due to its high $CO_2$ emissions (note that $N_2O$ is not included in the
present analysis as emissions are not included in the gridded CEDS and SSP database, but
would add a small contribution to the long-term impact of AGR). The second largest driver of
long-term temperature change is IND, demonstrating the importance of non-$CO_2$ emissions for
shaping relative weight over different time frames. Aviation is not included here, but was
recently evaluated by Lund et al. (2017).
The largest regional contribution to net near-term warming is caused by emissions in East Asia
(EAS) and North America (NAM), followed by South East Asia (SEA) and South Africa (SAF)
(Fig.2c). However, the relative contributions from individual species vary. In EAS and NAM,
as well as Europe (EUR), the impact of current emissions of cooling and warming SLCFs
approximately balance in the near-term and these regions cause comparable net warming
impacts on 10- and 100-year time scales, as seen by comparing the white and grey circles in
Fig. 2c. These balancing characteristics do not imply that SLCF emissions should not be

reduced, but that the net benefits on global temperature may be low if mitigation measures that simultaneously affect both cooling and warming SLFCs are implemented, in turn placing added focus on the need to reduce $CO_2$ in order to mitigate warming in both the near- and long-term. In SEA, SAF and South and Central America (SAM and MCA) methane and BC emissions are presently high while emissions of $CO_2$ and cooling aerosols are low compared to other regions, resulting in a net warming impact after 10 years that is substantially higher than that of $CO_2$ alone. This, in turn, suggest that using SLCF emission reduction to limit near-term warming would be more effective here than in many other regions. Such detailed characteristics at the emission source level are needed for the design of effective mitigation strategies.

Breaking the temperature impacts further down into economic sectors within each region (not shown), we find that the results largely mirror the relative role of species and sectors on the global level in Fig. 2b. The warming contributions in South America and Africa, and hence higher potential for net temperature reductions, stem primarily from the agriculture, waste management, and energy production sectors. In SAF, mitigation of BC emissions from the residential and transport sectors also play an important role. In most regions, emissions from IND cause a net negative impact on global temperature change, while in the ENE sector, impacts of cooling and warming SLFCs compete and warming from $CO_2$ is a key driver of both near- and long-term warming.

Overall, the potential for global temperature reductions inherent in the present SLCF emissions is highly inhomogeneous, and co-emitted species – including $CO_2$ – must be taken into account in any targeted climate policy for reduction of near-term warming. We emphasize that mitigation of SLCFs, while important, need to be sustained and complimentary to strong cuts in $CO_2$ for long-term reduction in global warming.

## 3.2 Temperature response to example mitigation measures

The results above suggest that strategies for emission reductions clearly can play out very differently in terms of net impact on global temperature across source region and sector. To further illustrate the importance of considering co-emissions and demonstrate the applicability of our dataset, we calculate the effect on global temperature in the near- and long-term following simplified examples of emission reduction packages in three of the global sectors (ENE, AGR and SHP). The measures are broadly assumed to be motivated by either *i)* air quality improvements (package 1, P1), *ii)* methane reductions (as part of the SDG agenda or climate mitigation) (P2) or *iii)* $CO_2$ reductions/climate targets (P3). Table 2 shows the set of species reduced in each case, with the percentage reduction given in parentheses. We note that these reductions are based on expert judgement given underlying assumptions, e.g., for the reduction in shipping speed, and are associated with uncertainties. Furthermore, they are assumed to occur instantaneously. However, as the primary purpose here is illustrative, the examples are kept idealized and should be interpreted as such.

The global temperature effect resulting from elimination of emissions in each package on 10- and 100-year time horizons is shown in Fig.3. The energy sector can be sub-divided into fossil fuel production/distribution and combustion categories. An air quality-driven set of measures

(P1), e.g., end-of-pipe measures such as scrubbers, filters and catalysts, could therefore be
implemented that would strongly reduce $SO_2$ and NOx emissions but not noticeably affect the
key methane or $CO_2$ contribution. Such measures are well understood, i.e., their efficiencies,
costs, and technical implementation has been well documented and real-life application is
already widespread but there is still large potential, especially in fast-growing economies. As
shown by the top bar on the left in Fig.3, the subsequent near-term temperature impact would
be a warming contribution due to removal of cooling aerosols, adding to the already large net
warming impact of the sector (Fig. 2b). As seen from the right-hand side of Fig. 3, the long-
term effect would also be minor, leaving the dominating $CO_2$ warming. A significant fraction
of methane emissions, originating from the production and distribution of fossil fuels, could be
mitigated separately from several other SLCFs, for instance by addressing venting and leaks
from oil, gas and coal exploration, and upstream and downstream gas flaring. Respective
measures would include capture/recovery and use of gas, as well as reduced and improved
flaring, with added benefits in terms of reduced $CO_2$ and BC (P2). This results in a notable
reduction in the near-term impact of the sector. Finally, P3 shows the impact of a dedicated
climate strategy, here illustrated by the difference between a middle-of-the-road and a below-
two-degrees scenario (in 2050, obtained from the GAINS model (Klimont et al., 2017)), where
more substantial $CO_2$ mitigation also result in larger reduction of the sector's long-term
temperature impact than in P2.
Due to the dominating contribution from methane to the temperature impact of the agriculture
sector, measures that primarily target other emissions, such as improving nitrogen use
efficiency (P1), unsurprisingly bring low net climate benefits unless accompanied by
simultaneous measures for methane reductions (P2). Examples of the latter is promoting dietary
changes, leading to lower meat consumption and consequently lower livestock numbers.
Reducing $NH_3$ and $NO_x$ (P1) could, however, bring important local air quality benefits, and our
results suggest that these would come with relatively small trade-offs from unmasking of
aerosol cooling, at least in terms of global mean temperature on this time scale. Only small
additional benefits (at a global scale) were estimated for the increased use of biogas (P3) based
in utilization of livestock manures. The net impact of the shipping sector (SHP) is a cooling in
the near-term, as shown in several previous studies (e.g., Berntsen & Fuglestvedt, 2008;
Fuglestvedt et al., 2009). Measures that eliminate shipping emissions of $SO_2$ (low sulfur fuels,
scrubbers) and NOx (selective catalytic reduction) hence result in an added near-term warming,
also when simultaneous elimination of the sector's $CO_2$ emissions occur (P2, P3).
This example is simplified and illustrative, and we calculate pulse-based temperature impacts
following instantaneous emission reductions. However, since our pulse-based emission metrics
can easily be used to study changes over time to any emission or policy scenario through
convolution (Aamaas et al., 2013), our dataset has broad applicability. In the next section, we
use precisely this method to quantify the impact of temporally evolving emissions according to
the most recent scenarios.

### 3.3 Temperature response to SLCFs and $CO_2$ under the SSP-RCP scenarios

While knowledge of the present-day emission composition and net temperature impact over
time is essential to support mitigation design and implementation, real-world emissions will

evolve following a combination of socioeconomic developments, technological advancement and policy adoption. Next, we investigate plausible pathways for the future impact of SLCFs and $CO_2$ by quantifying the global temperature change over the period 1900-2100 to regional and sectoral emissions following the SSP-RCP scenarios. In the following paragraphs, we show results from four of the nine SSP-RCP scenarios used in the present analysis (SSP1-1.9, SSP2-4.5, SSP3-7.0 and SSP5-8.5). These span the range of future emission evolutions, but we recognize that the realism of SSP5-8.5 is debated in the literature due to its very high emissions (e.g., Ritchie & Dowlatabadi, 2017).

Figure 4 shows the evolution of temperature response under the SSP-RCPs for our source regions, with corresponding results for the global economic sectors given in Fig. S3. Our emissions regions not only have large differences in terms of present-day emissions, but also of past evolution. This historical contribution, which was not captured in the analysis of the first half of the paper, brings NAM and EUR as the two largest contributors to the present-day warming (Fig. 4a) due to their much higher past $CO_2$ emissions, in line with previous literature (Höhne et al., 2011; Skeie et al., 2017). While presently being the largest emission source, EAS only surpasses EUR and NAM, in terms of contribution to temperature change, between 2020 and 2030 when the cumulative effect of $CO_2$ is accounted for. In SSP1-1.9, where emissions of $CO_2$ decline strongly during the first half of the century in all regions, the net temperature response levels off or starts to decline in the second half of the century. In the remaining scenarios, the net temperature impact increases over the century for all regions. EAS remains the largest contributor, whereas in SSP5-8.5 SAS overtakes NAM as the second most important region by 2100 and SAF reaches the same order of magnitude as EUR. This shows a projected shift in emissions and increasing importance of the developing world. We note that since our primary focus here is on quantifying the contributions to, and potential for further reduction of, near- and long-term temperature impacts, we do not include negative $CO_2$ emissions which is already a mitigation measure. Furthermore, the gridded SSP-RCP inventory only provide negative $CO_2$ as a separate category without information for mapping these emissions to economic sectors. We do, however, include the negative $CO_2$ category in our inventory of regional scenarios for further analyses beyond our study (see "Data Availability").

In our calculations, the net temperature response to emissions from the global energy (ENE) sector becomes larger than that due to AGR and RES in the early 2000s (Fig. S1a), after which ENE remains the largest individual sector until 2100 in all scenarios. The relative importance of AGR and ENE historically is yet another example of how including SLCFs can change relevance over different time frames, as also demonstrated by Reisinger and Clark (2018) for non-$CO_2$ livestock emissions. In our results, both the warming due to $CH_4$ from AGR and the contributions from cooling emissions from ENE act to shape the relative role of the two sectors over time. The global mean temperature impact of IND switches from a net cooling to a net warming in the late 20[th] century as the warming due to $CO_2$ accumulates and overwhelms the cooling from $SO_2$.

While the contribution from $CO_2$ to the net warming becomes dominant by 2100 for most regions and sectors in all scenarios, the relative importance of SLCFs and $CO_2$ continue to be highly variable across emission source over time, in particular under SSP3-7.0 and SSP5-8.5.

This can be seen in Fig.4b, where we break down the future net temperature response in 2030, 2050 and 2100 into individual contributions from methane, $CO_2$, BC and the sum of $SO_2$ and NOx. Here we show a selection of the source regions that differ notably in composition and temporal trend. See Fig. S4 for remaining regions.

The SSP-RCPs differ in both climate forcing targets and stringency of air pollution control, as well as underlying socioeconomic development. SSP1-1.9 is characterized by low societal challenges to mitigation and adaptation, and strong climate and air quality policies, resulting in rapidly declining emissions of both SLCFs and $CO_2$. However, even for strong air pollution there is a differentiation between high-, medium- and low-income countries, with a substantial time lag in the latter two (Rao et al., 2017). For example, emissions of $SO_2$ in SAS and SAF decline less than in other regions, subsequently maintaining a significant cooling contribution to the temperature impact. In the intermediate scenario, SSP2-4.5, there is a reduction in emissions, but this is delayed and slower compared to SSP1-1.9. In SSP3-7.0, the world follows a path with more inequality and conflict, where only weak air pollution control is implemented and the end-of-century climate forcing, and hence $CO_2$ emissions, is higher. Subsequently, emission trends and SLCF contributions display more regional heterogeneity. There is a particularly strong projected increase in methane emission in South Asia, Africa and South America in this scenario. While previous decades have seen a southeastward shift in air pollution emissions, from high income regions at northern latitudes to East and South Asia, these findings suggest that a second shift may be underway, towards low- and middle-income countries in the developing world. Further studies are needed to improve the knowledge about the resulting climate and environmental consequences, as well as how to strengthen the mitigation options, in these regions. While EAS remains the region with the largest warming impact by 2100 in all scenarios, the contributions to warming from methane and BC in SAF and SAS surpasses those of EAS in 2100 in both SSP3-7.0 and SSP5-8.5. The net temperature response to emissions in SAS increases from close to zero to a significant warming as $CO_2$ emissions increase. SSP5-8.5 is characterized by high challenges to mitigation and high climate forcing in 2100, but still assumes strong air pollution control since the high use of fossil fuels would otherwise result in unbearable air pollution levels. Combined, this leads to increasing temperature impact due to increasing $CO_2$ emissions, but lower SLCF impacts than in SSP3-7.0, but with a non-negligible contribution from methane for several regions. Hence, in medium- and low-income regions, SLCFs, and in particular methane, are projected to play a continued important role for future temperature change.

Clearly, and as expected, the largest difference in SLCF contributions to future temperature response is between SSP1-1.9 and SSP3-7.0. To see where the largest additional climatic benefit can be gained from mitigating SLCF emissions in line with SSP1-1.9, relative to SSP3-7.0, we show the difference in temperature between these two scenarios in 2030, 2050 and 2100 in Fig.5. Results are shown by region and sector, for all combinations where the temperature difference is greater than ±0.01°C. For comparison, the CMIP6 mean difference in projected surface temperature between SSP3-7.0 and SSP1-2.6 (which is close to SSP1-1.9 in emissions) is around 0.5 °C in 2050 and 2 °C in 2100 when accounting for all global emissions (Tokarska et al., 2020). As seen from Fig. 4 and Fig. S3, $CO_2$ is the key driver of this long-term temperature difference between the scenarios for most sectors and regions. However, as seen in Fig.5, there

are also important SLCF contributions, most notably from the large sources of methane;
agriculture, energy and waste management. Furthermore, 9 of the 12 top contributions are from
regions in Africa, South Asia or South and Central America, again demonstrating the
importance of the development in low- and middle-income countries for future levels of SLCFs.
Fig.5. also shows how the strong SLCF mitigation in SSP1-1.9, relative to SSP3-7.0, can result
in a net warming contribution to climate for some region-sector combinations, as exemplified
by the industry sector in East and South Asia. As shown by the panel on the right-hand side of
Fig. 5, for most sector/region combinations, around 10% of the avoided (or added) warming
from strong mitigation would be realized already by 2030, and around 40-50% by 2050.

## 491     4 Discussion

In terms of avoided global warming, there is much to be gained by moving from a global
emission pathway following SSP3-7.0 to one following SSP1-1.9, including contributions from
reductions of SLCFs, as discussed above. While a comprehensive assessment of policy and
technological interventions required to translate this potential to actual emission cuts is beyond
the scope of the present study, we outline key general features and discuss specific examples in
the case of methane, in the following paragraphs.

Available literature suggest that  rapid reductions of air pollutants' emissions are technically
possible drawing on experience in both developed and developing countries (Crippa et al.,
2016; Kanaya et al., 2020; Klimont et al., 2017) but would require simultaneous strengthening
of institutions to enforce the laws. The focus of policies would differ between OECD countries
and the developing world. As demonstrated by our findings, further measures in the OECD
would primarily focus on reducing emissions from residential heating, non-road transportation,
and agriculture while assuring enforcement of legislation in power and industry sectors. The
rapidly industrializing and developing countries would need to further strengthen legislation for
the power, industry, transport sectors, implement improved measures to reduce waste
management emissions, reduce emissions from agriculture, and provide wide access to clean
fuels securing cooking and heating needs. Several of these policies would contribute positively
to the SDGs (Rafaj et al., 2018). For methane, the non-$CO_2$ component found here to be most
important for future warming, reducing venting and increasing utilization of associated
petroleum gas in oil and gas exploration and increased use of biogas from waste should be a
priority, and the technical potential for considerable reductions until 2050 exists (Höglund-
Isaksson et al., 2020). Integrated response options that can deliver significant mitigation also
exist for the agriculture sector, including increased productivity of land used for food
production and improved livestock management (Smith et al., 2019). A similar suite of methane
measures is needed as for the developed and developing world, although waste management
requires larger transformation and there is additional significant potential to reduce emissions
from coal mining sector in the latter. A recent study suggests that anthropogenic fossil methane
emissions may be significantly underestimated (Hmiel et al., 2020), and as such, reductions
may be even more critical. Specific measures for reducing aerosols and ozone precursors in
order to improve air quality while contributing to climate change mitigation have recently been

assessed for South East Asia (UNEP, 2019) and Latin America (UNEP, 2018). As shown in the present analysis, contributions from SLCFs to temperature change are projected to increase strongly in the Middle East and Africa in several scenarios. An increasing carbonization in Africa south of the Sahara, primarily due to the increasing use of oil in the transport sector, has already been observed (Steckel et al., 2019). This underlines the need for further focus on these regions in future studies and assessments.

SSP3-7.0 and SSP1-1.9 not only differ in the stringency of the assumed air pollution control, but also in socioeconomic development and end-of-century climate forcing. To isolate the role of air pollution policies in the transition to a low warming pathway, a companion scenario to SSP3-7.0 has been developed, the SSP3-lowNTCF (Gidden et al., 2019). Here, the socioeconomic narrative is the same, but emission factors for the short-lived species are assumed to be in line with those in SSP1-1.9. The result is similar global $CO_2$ emissions but up to 60% reductions in global SLCF emissions in SSP3-lowNTCF relative to SSP3-7.0. Using the SSP3-lowNTCF emissions as input, we find that this in turn leads to a net temperature response to total global emissions in 2100 that is 13% lower in SSP3-LowNTCF than in SSP3-7.0 (an absolute difference of 0.5°C, from 3.7°C to 3.2°C in our calculations). For comparison, the net temperature response is 71% (or 2.6°C) lower in SSP1-1.9 than in SSP3-7.0 in our calculations.

The potential for reducing near-term warming by targeting BC emissions in the transport and residential sectors has been highlighted earlier (e.g., UNEP, 2011). We also find notable BC warming contributions from the residential sector in some regions, mainly South Asia and Africa, but estimate quite low BC effects from the transport sector. This has three main reasons. Firstly, since earlier studies (done about 10 years ago) there have been significant changes in legislation, and new diesel trucks and cars are (in several regions) equipped with particulate filters effectively removing BC. By now these vehicles represent a significant part of the fleet in many regions and the trend is expected to continue. Secondly, as described in Sect.2, we use an AGTP for BC that is 15% lower than in previous studies using the same methodology. This is done to account for the rapid adjustments associated with BC short-wave absorption (Stjern et al., 2017), which has been found to reduce the effective RF in a range of global climate models via changes in stability and cloud formation (Smith et al., 2018). For our study, this factor applies to BC emissions from all sources and hence results in a reduced the net warming impact. Finally, we account for cooling from nitrate aerosols from emissions of NOx, for which the transport sector is a significant source, even in regions where stricter vehicle emission standards (e.g., Euro 5) have been adopted.

4.1 Caveats and uncertainties

The AGTP is a well-established framework that has been applied in several studies of attribution of temperature impacts to emission sources and scenarios (e.g., Collins et al., 2013; Lund et al., 2017; Sand et al., 2015; Stohl et al., 2015; Aamaas et al., 2019). Here we have also consistently included the carbon-climate feedback in the AGTP for all species. This increases

the non-$CO_2$ AGTPs, however, less than initially suggested by Myhre et al. (2013) as discussed by Gasser et al. (2017). Figure S5 shows the global mean net temperature response to total emissions under 6 of the 9 SSP-RCPs, with and without the feedback. By the end of century, there is a 5-9% difference depending on scenario.

A key strength of the AGTP framework is that allows us to investigate the effects of individual species, sources and scenarios, which would be confounded by the low signal-to-noise ratio in fully coupled models, in a transparent manner. However, there are also caveats. Importantly, the AGTP metric is linear, while in reality the radiative efficiency can have non-linear dependencies on the background atmospheric conditions. In this study, we account for one part such non-linearities by using radiative efficiencies for the aerosols and ozone precursors that vary with emission location to calculate region-specific AGTPs. The part of the non-linearities caused by changing background levels of pollutants over time is, however, not included. For the well-mixed greenhouse gases $CO_2$, $CH_4$ and $N_2O$, the radiative efficiency (RE) is reduced with increasing atmospheric background concentrations. Previous literature suggests that the sensitivity to emission scenario is small, and the relationship between emissions and temperature response more linear, for $CO_2$ (Caldeira & Kasting, 1993). However, the same has not been shown for methane (and $N_2O$ – which is not considered here). We therefore perform an additional sensitivity test where we calculate an AGTP(t) that is adjusted to the global atmospheric concentrations over time (using the equation from Etminan et al. (2016) and global concentrations for each SSP-RCP from the IIASA SSP database (IIASA, 2020; Riahi et al., 2017)). Figure S5 shows the resulting temperature response, compared to the temperature response calculated with and without the CCf. As expected, using a dynamically adjusted RE results in a lower warming in the high emission scenarios and a slightly higher temperature response under low emissions. In the case of extreme scenario SSP5-8.5, the effect is of the same order of magnitude as that from adding the CCf, but of opposite sign. For aerosols and ozone precursors, potential saturation effects involve complex, spatially heterogeneous chemistry, cloud and climate interactions that require detailed chemistry-climate simulations to be resolved, and even then, may not be fully captured due to e.g., the coarse resolution of current models. We emphasize that the absolute magnitude of temperature changes quantified with the AGTP framework should be interpreted with care, as this method is primarily designed to study relative importance and relationships between individual emissions and sources.

Our analysis reflects best estimate input data to the extent possible, but results have considerable uncertainty, in emissions, RF and climate sensitivity. As shown in Fig. 2a, we estimate, due to uncertainty in RF alone, a 1 standard deviation range in the total net temperature response on the 10-year time horizon of ±0.01°C, about 38% of the net temperature response of 0.03°C (the range is considerably lower on the 100-year time scale as the RF of SLCFs is much more uncertain than that of $CO_2$). Uncertainties in emission inventories are difficult to quantify, but are generally considered lowest for $CO_2$ and $SO_2$ emissions, and high for carbonaceous aerosols (Hoesly et al., 2018). The level of uncertainty also differs across regions and sectors, with emissions from nature related emissions (e.g., agriculture, landfills) more uncertain than emissions in the fossil-fuel sector (Amann et al., 2013; Jonas et al., 2019). Moreover, recent studies point to emission trends that are not accurately represented in the global inventory, such as $SO_2$ and NOx in China (Zheng et al., 2018) and fossil fuel $CH_4$

emissions (Hmiel et al., 2020). However, due to high spatiotemporal variability and lack of
consistent data, a comprehensive uncertainty analysis at the regional and sectoral level is
challenging. The equilibrium climate sensitivity (ECS) inherent in the climate response in IRF
used in the present analysis is 0.885 K $(\text{Wm}^{-2})^{-1}$. This is in the upper range reported by Bindoff
et al. (2013), but lower than many recent estimates (Forster et al., 2019; Zelinka et al., 2020).
While emission uncertainties can have a strong spatiotemporal character, changes in the ECS
mostly act to scale estimates for all sectors and regions but is less important for their relative
ranking.
Our analysis is limited to temperature change as a measure of climate impacts. SLCFs, and in
particular aerosols, also play a key role in shaping local and regional hydrology and dynamics.
Comparing the SSP3-7.0 and SSP3-lowNTCF scenarios, Allen et al. (2020) recently found a
significant precipitation increase due to removal of aerosols, with the strongest moistening
trends over Asia. An increase in the Asian summer monsoon precipitation in scenarios with
strong air pollution reductions was also recently found by Wilcox et al. (2020). Hence, further
studies using coupled models are needed to fully capture the effects of the SLCFs under SSPs
on local climate and environment.

## 5 Conclusions

Complimentary mitigation of $CO_2$ and other LLGHG with SLCFs is of key importance for
achieving the ambitions of the Paris Agreement and meeting the Sustainable Development
Goals. Using the concept of Absolute Global Temperature change Potential (AGTP), an
emission metric-based emulator of the climate response, we here investigate the contribution
from emissions of SLCFs and $CO_2$ from 7 economic sectors in 13 source regions to global
temperature change. In addition to quantifying the near- and long-term temperature response to
present-day emissions, i.e., in line with the traditional emission metric studies, we evaluate the
role of individual SLCFs and $CO_2$ as projected by the most recent generation scenarios, the
Shared Socioeconomic Pathways (SSPs), with greater regional and sectoral detail than previous
literature. We account for the geographical dependence of the radiative forcing of SLCF
emissions, as well as the current understanding of global-scale indirect and semi-direct aerosol
forcing. A key update to our method relative to the bulk of comparable literature, is a treatment
of the carbon-climate feedback in the AGTPs of the SLCFs.

As is well established, $CO_2$ is the dominant driver of warming on longer time scales and any
strategy for limiting long-term temperature change critically depends on deep cuts in $CO_2$
emission. As shown by our results, $CO_2$ also give a significant contribution to near-term
warming. The potential for additional reductions in near-term temperature change from
reductions in present-day SLCF emissions is highly inhomogeneous across region and sector.
Key in all regions are the major emitters of methane, in particular agriculture and waste
management, but also energy production. In contrast, some sectors and regions, notably
industry, energy and transport in East and South Asia and the Middle East, have strong
contributions from cooling SLCFs resulting in a net negative near-term temperature impact or
an approximate balance between cooling and warming SLCFs. While this does not imply that
mitigation measures should not be implemented, understanding of the detailed characteristics
and relevance over time at the emission source level is needed for the design and assessment of
mitigation strategies.
The regional heterogeneity in SLCF emissions and subsequent contributions to global
temperature change continues under most of the nine SSP-RCP scenarios considered here.
While $CO_2$ becomes the dominant contributor to warming in all regions over time, SLCFs are
projected to continue to play an important role for global temperature change over the 21$^{st}$
century in many regions. In particular, emissions of SLCFs in East and South Asia are projected
to remain high, at least until the mid-21$^{st}$ century. Moreover, there is a shift in emissions towards
low- and middle-income countries in the developing world. Notably, a strong increase in
emissions in Africa south of the Sahara is projected under most of the SSP-RCPs considered,
and is especially pronounced in SSP3-7.0 and SSP5-8.5. Hence, in addition to the focus on the
current  major current sources of SLCFs, enabling technological and legislative development
on the African continent will likely be of key importance for a transition from high emission
pathways towards one in line with SSP1-1.9 and the ambitions of the Paris Agreement, which
in turn could give reductions in global warming already over the next couple of decades.
Technological advancement could bring benefits even if there is no dedicated climate policy
addressing SLCFs, simply by reduced emission factors, as demonstrated by the SSP3-lowNTCF
scenario.
The large spatiotemporal heterogeneity in emissions trends and subsequent temperature
responses underlines the need to go beyond global emission scenarios. By quantifying the
global temperature response to emissions from 13 regions, 7 sectors and 9 scenarios in a
consistent and transparent framework, we provide a more comprehensive dataset than, to our
knowledge, currently exists. We note that the AGTP framework is primarily designed to study
relative importance and relationships between individual emissions and sources, and that the
absolute magnitude of temperature responses should be interpreted with care due to the linearity
of the AGTP. The uncertainties in emissions could also affect the regional and sectoral ranking
but are poorly known. However, by making our full dataset publicly available, we provide a
tool that enables further analysis and comparison of e.g., mitigation strategies at the sectoral
and regional level without the use of complex models.


**Data availability**

Regional and sectoral emission timeseries, AGTPs and temperature responses are publicly
available via Figshare under the DOI https://doi.org/10.6084/m9.figshare.11386455 (Lund et
al., 2020). The full set of gridded SSP anthropogenic emission data are available from the
ESGF system (https://esgf-node.llnl.gov/search/input4mips/, Cinquini et al. (2014), last
access: December 2019). Code available upon request from Marianne T. Lund
(m.t.lund@cicero.oslo.no).

**Author contributions**

Lund led the study, prepared the input data and wrote the paper. Aamaas performed the emission metric and uncertainty calculations. Stjern and Samset produced the graphics. Klimont and Berntsen contributed to the design of the analysis. All authors contributed to the manuscript preparation.

**Competing interests**

The authors declare that they have no competing interests.

**Acknowledgements**

The authors acknowledge funding from the Research Council Norway grant no. 248834 (QUISARC). We thank Glen Peters and Robbie Andrews (CICERO) for assistance with the technical implementation of the carbon-climate feedback. We also thank the two anonymous referees for their comments and suggestions.

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

**Tables:**

*Table 1: Constants of the Geoffroy et al. (2013) IRF.*

|  | Mode 1 | Mode 2 |
|---|---|---|
| $c_j$ | 0.587 | 0.413 |
| $d_j$ (years) | 4.1 | 249 |

 *Table 2: Summary of species considered in the idealized emission reduction packages, the*
 *percentage reduction assumed, and example polices. All percentages refer the total emissions*
 *of a given sector, not total anthropogenic.*

| Sector | Package 1 (P1) | Package 2 (P2) | Package 3 (P3) |
|---|---|---|---|
| ENE [a] | End-of-pipe measures | Reduced loss in fossil fuel production and distribution | Climate strategy |
| | $SO_2$ (85%) $NOx$ (75%) | $CH_4$ (75%), BC (85%) $CO_2$ (3%) [b] | $CO_2$ (65%), $CH_4$ (40%) $SO_2$ (65%), $NOx$ (45%) BC (35%) |
| AGR | Nitrogen use efficiency and technical improvements | Meat reduction | Increase in biogas use |
| | $NH_3$ (65%) $NOx$ (60%) | $CH_4$ (35%) $NH_3$ (75%) $NOx$ (75%) | $CH_4$ (2%) $NH_3$ (10%) $CO_2$ (negligible) |
| SHP | Scrubbers and particulate filters | Slow-steaming [d] | Strong increase in LNG capacity |
| | $SO_2$ (95%) [c] $NOx$ (75%) BC (85%) | $CO_2$ (35%) $SO_2$, $NO_x$, (35%) BC (20%) | $CO_2$ (5%) $SO_2$, (90%) $NO_x$, (55%) BC (30%) |

 a) Here stationary combustion in power and industry.
 b) Through use of recovered CH4 instead of coal as fuel in oil, gas and coal industry.
 c) The reduction level is based on a year 2015 baseline with relatively high sulfur content for international
 shipping
 d) Assuming about 20% reduction in speed

**Figures:**

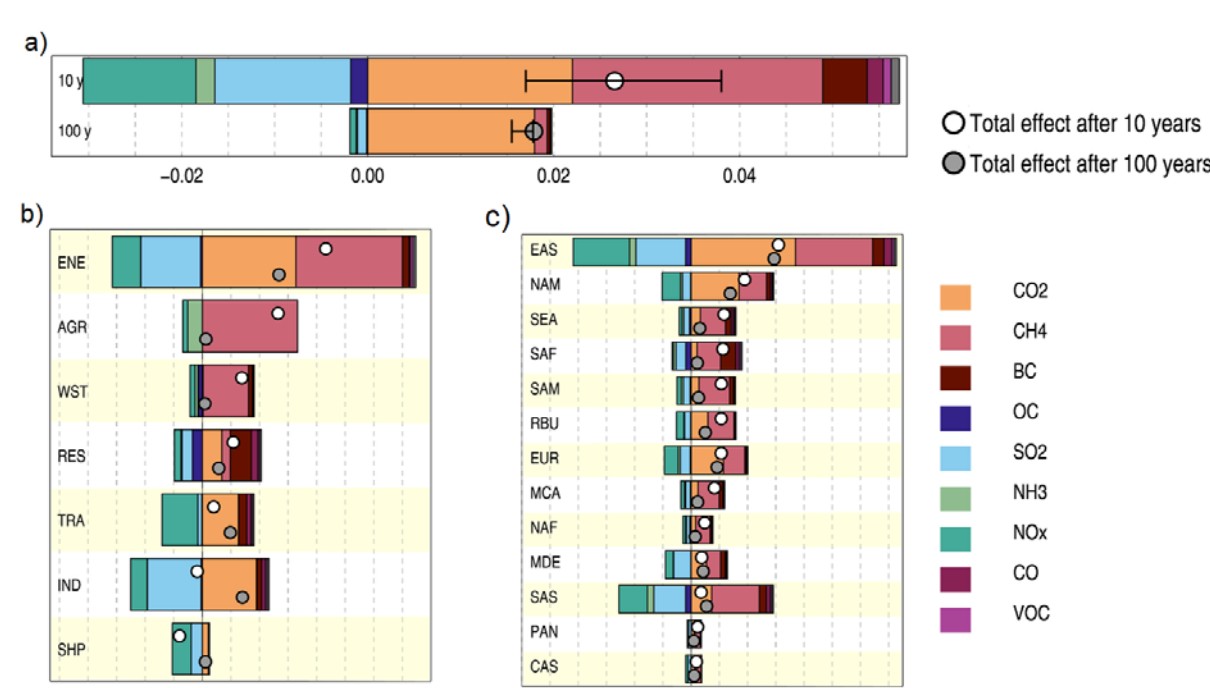

*Figure 1: Emission source regions and sectors used in the analysis.*

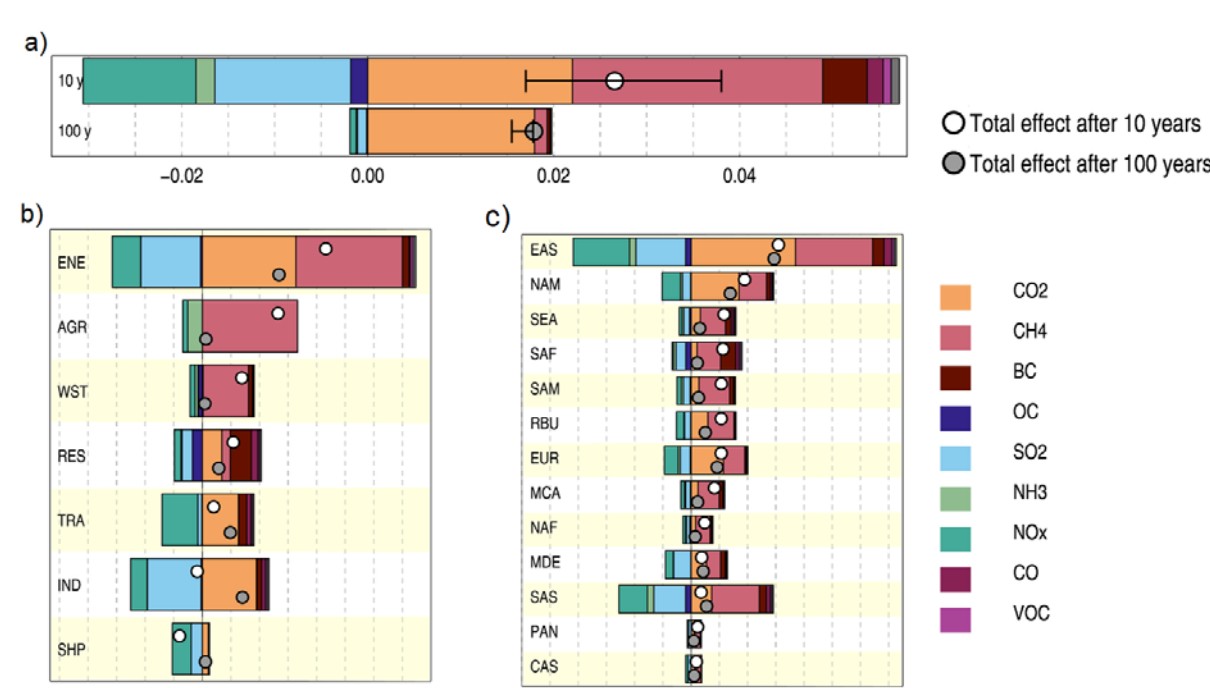

*Figure 2: Global-mean surface temperature impact 10 and 100 years after one year of present-day (i.e., year 2014) emissions of SLCFs and $CO_2$ for: a) global total emissions, b) emissions from seven major economic sectors, and c) total (i.e., sum of all sectors) emissions in 13 sources regions. Panels b and c are sorted by total net effect on the 10-year timescale (white circle). Error bars (±1 standard deviation) in the top panel represent the range in total net temperature impact due to uncertainties in radiative forcing.*

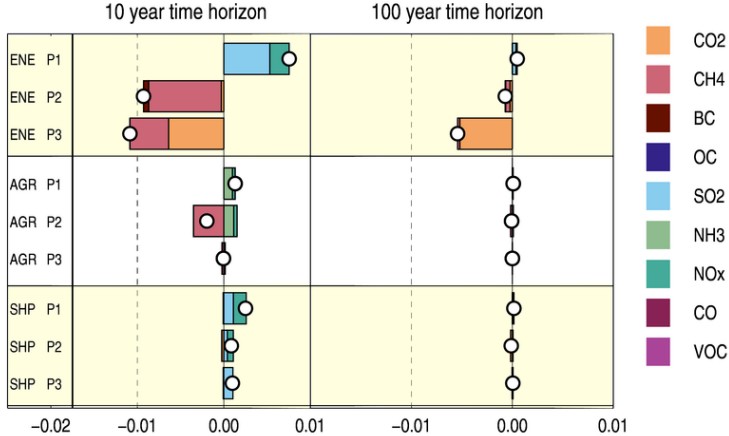


*Figure 3: Global-mean surface temperature impact on 10 and 100 year time horizons resulting*
*from instantaneous reductions of different sets (listed in Table 2) of SLCFs and CO₂ emissions.*
*White circles indicate the net impact of these reductions.*

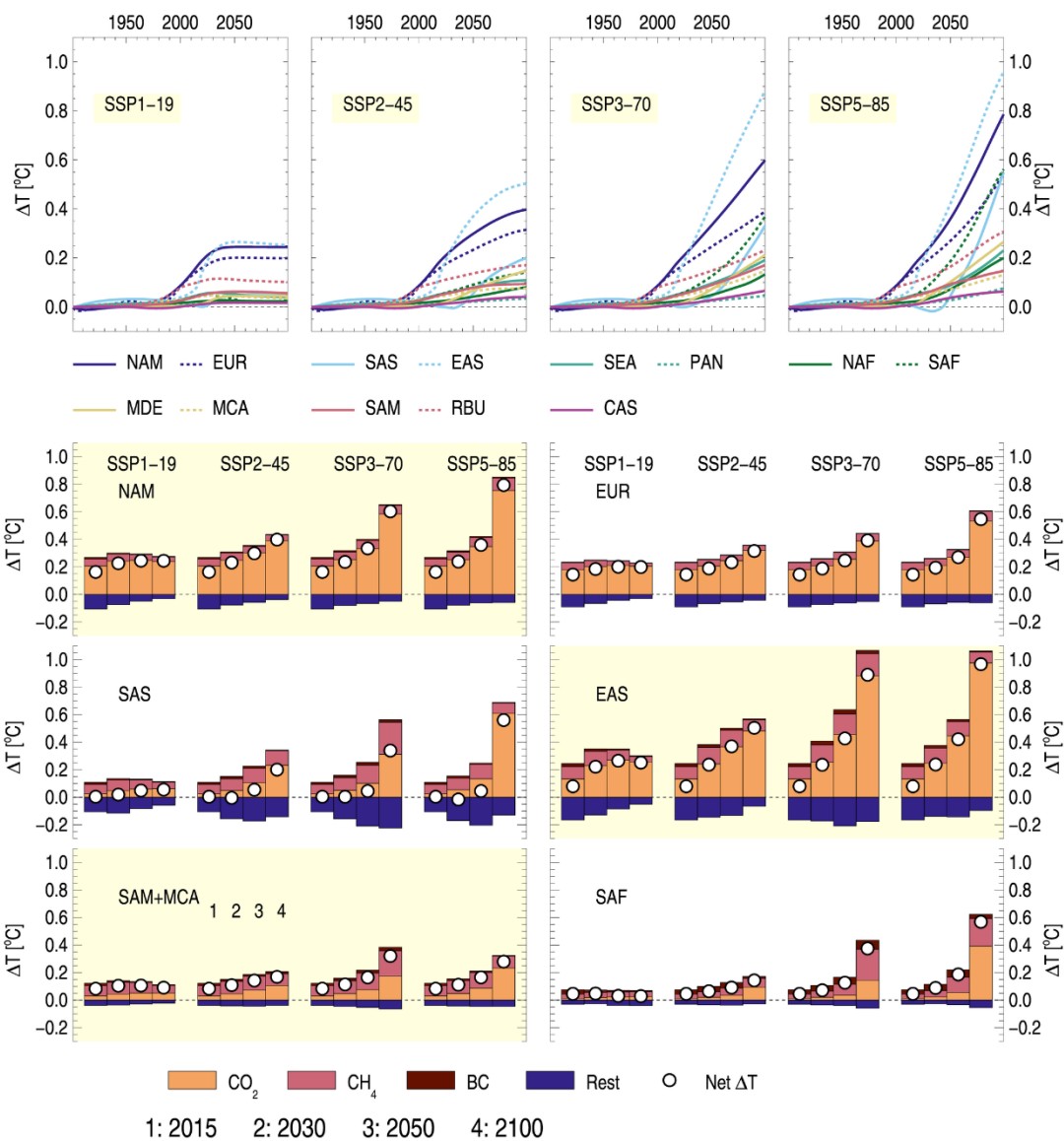


*Figure 4: Global mean temperature response to historical emissions and future SSP pathways:*
*a) Net (i.e., sum over all species and sectors) response over the period 1900 to 2100 for each*
*region and scenario and b) net response in 2015, 2030, 2050 and 2100 to emissions in six*
*regions broken down by contributions from $CO_2$, BC, methane and the sum of $SO_2$, OC, $NH_3$*
*and ozone precursors (i.e., "Rest").*







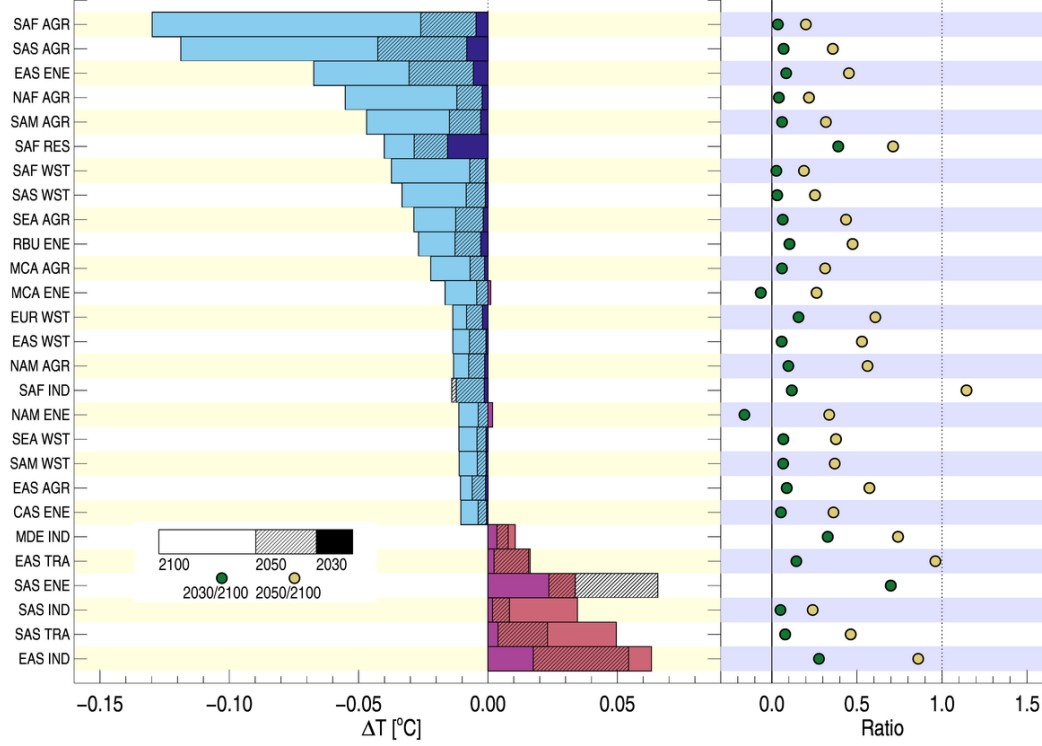


*Figure 5: Difference in net SLCF (i.e., sum of all components except CO₂) temperature*
*response between SSP1-1.9 and SSP3-7.0 in 2030, 2050 and 2100 by region and sector. Only*
*combinations of sectors and regions where the differences in global temperature response is*
*larger than ±0.01 °C are shown. For each of these combinations, the panel on the right shows*
*the ratio between the temperature response difference in 2030 and 2100 and between 2050 and*
*2100.*










