# Peer review of "A continued role of Short-Lived Climate Forcers under the Shared Socioeconomic Pathways Marianne T. Lund1\*, Borgar Aamaas1, Camilla W. Stjern1, Zbigniew Klimont2, Terje K. Berntsen1,3, Bjørn H. Samset1 1 CICERO, Center for Intern"

_Earth System Dynamics, 2020_

## Referee Comment (RC1) · Anonymous Referee #1 · 13 Apr 2020

GENERAL COMMENTS

The manuscript makes an important contribution to the literature by providing a detailed assessment of SLCF emissions, implications of mitigation approaches, and understanding the implications for global temperature over different time horizons and under different SSPs.

I have two major, related methodological concerns that I believe the authors need to address (but also should be able to address) for the paper to deliver on its promise. Both concern the use of AGTP and convolution of an IRF to derive outcomes over different time horizons and for emission pathways, and the fact that the exact methodology is

too opaque yet choices here are critical.

My first concern is that a comparison should be shown (can be done in Supplementary Material) of how the IRF and AGTP used in this paper compares to the IPCC AR5 and body of literature used in the draft IPCC AR6 (the authors obviously can't cite the IPCC AR6 draft, but it would be enormously helpful if their IRF and AGTP had a strong resemblance to what is coming out of the AR6 draft, because if it doesn't, it clear is missing some important science point).

One important aspect of this is the treatment of climate-carbon cycle feedbacks. There is enough literature and recommendations in various papers arguing that this should be included, and the consequences are non-trivial for SLCFs especially for longer time horizons of 100 years – based on the AR5, this more than doubles the AGTP100 of methane. Since the goal of the paper is to describe the impact of SLCF emissions and mitigation over both short and long time horizons, the choice here is critical – but I'm not at all clear based on the current manuscript what choice was made.

I'd argue strongly that the authors should include a climate-carbon cycle feedback in their IRF – as not doing so would make the results for 100-year horizons, and for emission pathways (i.e. the effect of sustained SLCF emissions) misleading. Given the different lifetimes within SLCFs, this could also affect the ranking of different regions and sectors – it would not be a uniform scaling such as from the choice of ECS. So this really matters in my view for the validity of findings.

I would therefore ask the authors to (a) make fully transparent how their IRF and AGTP compares to IRF and AGTP that include climate-carbon cycle feedbacks from the IPCC AR5, and glancing at the studies and assumptions used in the AR6 draft, and (b) if their current IRF and AGTP does not include climate carbon cycle feedbacks or is missing some other critical aspects, to update their IRF and re-run their analysis. I'm hoping that this would be possible without requiring too much additional work since the framework for analysis should not change (and some results may not change either –

which in itself would be a useful finding from this study!)

My second concern is that their IRF and AGTP apparently does not include saturation effects arising from concentration changes (although it took me until the discussion on page 12 to realise this, which underscores my sense that the methodology is not transparent enough). The use of a linear AGTP is not acceptable in my view for the part of the paper that compares outcomes under different SSPs and mitigation targets. For some gases (methane as the biggest forcer included), their concentration differs markedly between the stringent and non-mitigation scenarios, which has a substantial effect on their radiative efficacy and hence contribution to warming over time. It is simply not defensible in my view to exclude this dependency but in a paper that seeks to evaluate the contribution to temperature from different gases under those different scenarios. Using a dynamically updated AGTP (i.e. adjusted based on concentration of each gas) could well change some of the results substantially (at least sufficiently to make the quantitative results questionable).

Again, I think this is doable – it would not be hard to scale the AGTP based on the concentration of each gas and changing radiative efficacy, and re-run the analysis with such a dynamically updated AGTP. As for my other main comment, the framework for analysis would remain unchanged, and some or many key results may or may not change – which, again, would be a useful result in itself.

All other comments are comparatively minor (though some include requests to broaden discussion or restructure some sections), as detailed below.

SPECIFIC COMMENTS

L83: "increase" should come after "temperature"

L91: "complimentary" should be "complementary" (different meaning!)

L96: insert "sources and" before "mitigation strategies"

L98: "inexorably" is too strong: not all SLCFs are (especially HFCs, and methane not

in all regions)

L112: insert "co-emitted" after encompass; also, I feel it is not correct to claim that sulfate aerosols have received considerably less attention so far – certainly in the 1990s that was the dominant aerosol included in climate studies. This should be clarified a bit and some of the older literature may well be highly relevant here (e.g. focus in the US on sulfate reduction from energy systems).

L117-121: I can't agree with that generic claim: the SRES scenarios had a wide range of evolution of methane emissions, with significant continued increases in emissions especially in the A2 scenario but also A1FI. SSPs are more nuanced but there hasn't been a material shift (unless you focus only on aerosols here – in which case, say so).

L160-191: As per my main comment, please expand this methodological section (possibly using SM) to demonstrate how the IRF and AGTP used in this paper ompares to other IRFs. In particular, clarify whether longer-term warming contributions related to climate-carbon cycle feedbacks have been included (I argue strongly you should – tell us what the AGTP100 is for methane and HFC23). Also, add a comment here about how the AGTP adjusts over time in response to changing global GHG concentrations (again as per my main comments, I think it has to be changed dynamically to allow authors to derive conclusions about differences between SSPs/RCPs).

L216: this section is not well structured in my view. It makes it hard to derive clear conclusions. I would suggest to improve on the structure by having one discussion about sectors, and another one about regions; also ensure you add a long-term (100 year) dimension, at present most of the discussion is for the near-term horizon.

L218-223: I can see the benefits of using 10 years, but I also struggle with the claim that this is "commonly used". Especially if the authors accept my main comment, that they need to re-do their analysis with a revised IRF/AGTP, I would urge you to consider a 20-year time horizon. The reason is that (a) this is in fact commonly used (GWP20), but also (b) that 20 years puts us very close to the time when temperatures should

(begin to) peak in 1.5°C scenarios – so 20 years is much more policy relevant in my view than 10 years, which is really just the near-term rate of change.

L226/227: add a bit of nuance here: the lifetime of SLCFs varies widely, with some causing warming for many decades (methane) whereas for others the bulk of warming is in the space of a few years.

L261-277: there's a bit of confusion about whether "mitigation potential" refers to the potential to reduce the emissions of a given SLCF, or to the potential for an intervention that might affect a range of SLCFs to reduce or increase temperature in the near or long term. These are very different aspects. I would reserve the word "mitigation" for anything that focuses on the reduction of emissions of a given species, and from there discuss the implications of such actions for temperature once changes in emissions of co-emitted species are taken into account over different time frames.

L279: It would be really helpful if this section could clarify the scale of mitigation out-comes from SLCF mitigation compared to CO2 (and other long-lived GHG) mitigation. This would help keep the importance of SLCF mitigation in perspective, and allow the authors to use words such as "significant" with a lot more precise and justified mean-ing. If you only compare outcomes between SLCF mitigation approaches, but don't provide an overall scale (how much of the total mitigation in a given scenario comes from SLCFs, how much comes from CO2 and other LLGHGs), the paper could poten-tially be dancing on the head of a pin. You need to demonstrate how relevant this SLCF mitigation is in the bigger context (essentially a brief update from Shindell et al 2012).

Also, I feel this section needs to spell out in quite a bit more detail the assumptions behind each policy entry point and how this translates into quantified emission reduc-tions. E.g. L285/286 says that P2 is about methane reductions, but then L305/306 seems to suggest that it can also be about CO2 reduction in the energy sector? Also more details are needed to understand the detailed emission reductions, and chem-istry assumptions, for the agricultural mitigation scenarios (a lot of policies that target

agricultural methane will affect agricultural N2O within farm systems).

So I think the authors need to provide much more detail and quantification of how the broad policy principles in P1-P3 translate into mitigation of individual species for the different sectors. It's fine if there are subjective choices made – but we need to know what exactly those choices were to better understand to what extent the results are a function of those choices, or of the properties of the individual species that this paper helpfully aims to disentangle.

L317: add "and mitigation targets" or something like this to the section heading, as the scenarios explored are not just the SSPs but the imposition of different mitigation targets on the SSPs (i.e. they are SSPs plus climate policy). Also clarify whether the way that the mitigation of SLCFs is then implemented follows the SPA protocol developed for mitigation modelling using SSPs (Kriegler E, Edmonds J, Hallegatte S et al (2014) A new scenario framework for climate change research: the concept of shared climate policy assumptions. Climatic Change 122(3): 401-414), since this could well affect how individual SLCF emissions change for different regions.

L324: I question the utility of using SSP5-8.5 for this paper. This scenario has value but by now is clearly counterfactual as far as emissions are concerned. This would not be a critical issue, but at the same time the paper is missing a much more relevant scenario such as SSP2-2.6, or SSP5-2.6. As it stands, the only stringent mitigation scenario is for an SSP1 world, which is only one of many worlds, understanding how SLCF emissions might evolve in a different socio-economic context but also stringent mitigation would be much more valuable than to take up space for the largely academic SSP5-8.5 scenario. So my main concern is: add a stringent mitigation scenario (RCP2.6) using a different SSP (other than SSP1), otherwise this paper is missing a really important dimension. If you then keep the 8.5 scenario or drop it is in a way secondary.

L336/337: "we note that negative CO2 emissions are not included in these calculations": I'm puzzled by this. How can you evaluate SSP1-1.9 without negative emissions? Why not? This problem would only grow if he authors follow my advice to include SSP2 or SSP5-2.6.

L341/342: There seems to be a rather important finding buried here: are the authors saying that globally, energy contributed less to actual temperature change than agriculture and RES? If correct this might be worth highlighting more prominently to show how including SLCFs can change relevance over different time frames. Not that this should take away from the critical importance of mitigating CO2 from ENE, but it does seem a significant element. Another study that looked at warming attributable to livestock seems to go in a similar direction (Reisinger A, Clark H (2017) How much do direct livestock emissions actually contribute to global warming? Global Change Biology DOI: 10.1111/gcb.13975).

L376/377: I had to read this a few times to understand the "put another way" – might be worth rephrasing or disentangling a bit

L378-390: again here, as for section 3.2, I would like to see a comparison with mitigation achieved by CO2 reductions, simply to avoid readers to take away misleading conclusions that somehow SLCFs are the dominant issue for climate change – I would say they are an important but second-order issue. Useful if the paper could state and substantiate this in some way. Also for L393-395: there is "much" to be gained – how much? Compared to how much from CO2?

L395-422: I find this section weak on actual policy, and inconsistent: for some sectors, authors mention specific interventions, whereas for agriculture, it just says "addressing agriculture emissions" – that's not a policy or intervention. Expand this to illustrate consistently what feasible interventions are for all sectors (including a brief flag for supply vs demand side interventions).

L424-43: this is a useful thought experiment: how much warming would be avoided simply by improving technology for SLCFs (i.e. reducing emission factors consistent with SSP1), even in the absence of any dedicated climate policy (i.e. SSP3-7.0 vs

SSP3-lowNTCF).

L449-480: please break this discussion into chunks – lots of different issues being discussed in a single mammoth paragraph. As flagged in main comments, using non-dynamic AGTP to explore SSP/RCP pathways is a real problem that the authors have to address.

L464-466: agricultural non-CO2 emissions should be included in this list as they are also highly uncertain especially in developing regions (AFR, SEA, SAS).

L486: add that emission reductions of SLCFs have to be sustained to achieve long-term temperature change

L494-498: You could emphasise more strongly that this technological advancement brings benefits even if there is no dedicated climate policy addressing SLCFs, simply by reducing emission factors.

---

## Referee Comment (RC2) · Anonymous Referee #2 · 14 Apr 2020

The manuscript emphasizes the importance of SLCF agents, especially for the short-term impacts of climate scenarios, with some emphasis on methane. It is concluded that SLCFs continue to play a role in many regions. While it is important to reiterate this message, it is not so obvious what new findings are being presented. On several occasions, the results reinforce what is known, which does not justify publication.

The results for methane depend on methodological assumptions that are not transparent (e.g., emission categories) nor are they discussed in sufficient detail in the presentation of results. I found the discussion about the changing role of BC interesting, which could be highlighted more. I also recommend emphasizing regional differences more

strongly. The finding that SLCFs are particularly relevant for low- and medium-income countries is relevant. In general, it would be good to deepen such analyses and bring new aspects forward more clearly.

There are some rather bold simplifications in the treatment of aerosols; e.g., it is not clear how the radiative properties of partially absorbing aerosols (with BC) are accounted for. They sensitively determine the radiative cooling efficiency. NOx is mentioned on several occasions, but its role is unclear. How is nitrate been included? It is semi-volatile and responds to changes in sulfate and ammonium. Has that been accounted for? This is particularly relevant for the comparison of scenarios.

A relatively large temperature signal is expected from the indirect effects of aerosols on clouds, being highly non-linear especially at low pollution levels. I find the scaling by a factor of 2.1 to the impact of sulfate questionable. I recommend investigating (and showing) how sensitive the results are toward this assumption. There could be large regional differences,

l.173 mentions a lack of information. Can't you get this from the chemistry-transport model?

l.175: The description of the -15% for BC after l.175 is unclear (e.g., the rapid adjustment). Can you explain?

l.190: "lower than in the literature". By how much? By 0.885/1.06? Is the effect linear?

l.200: I am doubtful about the linearization of the temperature response by multiplying the emissions with the AGTPs. There are models available to compute this properly. This is particularly relevant for aerosols and ozone (the latter not being discussed at all), and to a lesser extent for methane, which has significant indirect effects, e.g., though ozone. Has this been accounted for?

l.210 Mentions ozone (also l.148), but it does not appear in the rest of the manuscript. It does not show in figures 2 and 3. Why has it not been included?

l.241: There is much debate about CH4 emissions from the fossil fuel sector. What has been assumed in the calculations, and how does it compare with recent estimates? Methane is emphasized in the conclusions, but the attribution of emissions to sectors is not transparent. It would be interesting to deepen the discussion about the role of methane. Currently, the results are being reported but not really analyzed.

l.261: This is an interesting result that could be explained and emphasized more strongly.

l.364-366: This is interesting and could be explained and emphasized more strongly.

l.443-445: This is interesting and could be explained and emphasized more strongly.

l.468-470: This is interesting and could be explained and emphasized more strongly.

---

## Author Comment (AC1) · 24 Jun 2020

Response to comments by anonymous referee #1 on "A continued role of Short-Lived Climate Forcers under the Shared Socioeconomic Pathways" by Lund et al.

We thank the referee for the detailed and thorough review of our paper, which has contributed to substantial improvements to our manuscript. Following the general comments and suggestions, we have repeated the analysis accounting for carbon-climate feedbacks and performed sensitivity tests to explore the impact of methodological choices, given in the supplementary material. We have also made substantial additions the Methods section, as well as changes to improve the flow of section 3.1. Responses to induvial comments are given below.

GENERAL COMMENTS
The manuscript makes an important contribution to the literature by providing a detailed assessment of SLCF emissions, implications of mitigation approaches, and understanding the implications for global temperature over different time horizons and under different SSPs. I have two major, related methodological concerns that I believe the authors need to address (but also should be able to address) for the paper to deliver on its promise. Both concern the use of AGTP and convolution of an IRF to derive outcomes over different time horizons and for emission pathways, and the fact that the exact methodology is too opaque yet choices here are critical.

My first concern is that a comparison should be shown (can be done in Supplementary Material) of how the IRF and AGTP used in this paper compares to the IPCC AR5 and body of literature used in the draft IPCC AR6 (the authors obviously can't cite the IPCC AR6 draft, but it would be enormously helpful if their IRF and AGTP had a strong resemblance to what is coming out of the AR6 draft, because if it doesn't, it clear is missing some important science point).

One important aspect of this is the treatment of climate-carbon cycle feedbacks. There is enough literature and recommendations in various papers arguing that this should be included, and the consequences are non-trivial for SLCFs especially for longer time horizons of 100 years – based on the AR5, this more than doubles the AGTP100 of methane. Since the goal of the paper is to describe the impact of SLCF emissions and mitigation over both short and long time horizons, the choice here is critical – but I'm not at all clear based on the current manuscript what choice was made.

I'd argue strongly that the authors should include a climate-carbon cycle feedback in their IRF – as not doing so would make the results for 100-year horizons, and for emission pathways (i.e. the effect of sustained SLCF emissions) misleading. Given the different lifetimes within SLCFs, this could also affect the ranking of different regions and sectors – it would not be a uniform scaling such as from the choice of ECS. So this really matters in my view for the validity of findings.

I would therefore ask the authors to (a) make fully transparent how their IRF and AGTP compares to IRF and AGTP that include climate-carbon cycle feedbacks from the IPCC AR5, and glancing at the studies and assumptions used in the AR6 draft, and (b) if their current IRF and AGTP does not include climate carbon cycle feedbacks or is missing some other critical aspects, to update their IRF and re-run their analysis. I'm hoping that this would be possible without requiring too much additional work since the framework for analysis should not change (and some results may not change either – which in itself would be a useful finding from this study!)

   a)   In the present analysis we do not report normalized metrics, have different geographical definitions than those used in IPCC AR5 (and other literature), and include various small

updates compared to IPCC AR5 (e.g., radiative efficiencies calculated using Etminan et al. (2016) , which makes a direct comparison difficult. However, the reviewer raises a fair point as our results can readily be used to present new GTPs. To assess the order of magnitude difference that may arise from these methodological choices, we have repeated our AGTP calculations and pulse-based analysis using different combinations of carbon dioxide and climate response IRFs from the literature. A comparison of selected AGTP timeseries, as well as examples of how GTPs and temperature responses to individual species are affected, is presented in the supplementary material.

Specifically, we use the Joos et al. (2013) CO2 IRF with Boucher and Reddy (2008) (as in AR5), Gregory et al. (2013) (as in the rest of our study), and Gasser et al. (2017) temperature IRFs. Additionally, we add two runs where we compare results using the CO2 IRFs with and without carbon climate feedback from Gasser et al. (2017). The most notable differences arise from the switch from Boucher and Reddy (2008) $IRF_T$ to Gregory et al. (2013) or Gasser et al. (2017). We also note that the sign of the difference (i.e., lower/higher values) depend on time horizon. The overall picture of our findings does not change, but the sensitivity analysis is a useful documentation.

Finally, the manuscript has been updated with more clear descriptions of methodological choices, including the use of Etminan et al. (2016) radiative efficiency equations, choice of IRFs and treatment of carbon-climate feedback (see below).

b) We thank the review for raising the point about climate-carbon cycle feedback (CCf). This is an important aspect but was neglected in our first calculations. We have now included the CCf using the framework developed by Gasser et al. (2017) with the OSCAR v2.2 simple earth system model, updating all figures and results. Since there are other approaches to accounting for CCf in the literature, we also provide AGTPs both with and without the CCf included in the data repository. As discussed in Gasser et al. (2017), the addition of a CCf term according to their approach increases the non-CO2 metrics, but less so than initially suggested by IPCC AR5 using the more simplified Collins et al. (2013) approach. This increase does not alter the overall picture and conclusions from our analysis. Nevertheless, the consistent treatment of CCf is a significant improvement to our paper.

My second concern is that their IRF and AGTP apparently does not include saturation effects arising from concentration changes (although it took me until the discussion on page 12 to realise this, which underscores my sense that the methodology is not transparent enough). The use of a linear AGTP is not acceptable in my view for the part of the paper that compares outcomes under different SSPs and mitigation targets. For some gases (methane as the biggest forcer included), their concentration differs markedly between the stringent and non-mitigation scenarios, which has a substantial effect on their radiative efficacy and hence contribution to warming over time. It is simply not defensible in my view to exclude this dependency but in a paper that seeks to evaluate the contribution to temperature from different gases under those different scenarios. Using a dynamically updated AGTP (i.e. adjusted based on concentration of each gas) could well change some of the results substantially (at least sufficiently to make the quantitative results questionable). Again, I think this is doable – it would not be hard to scale the AGTP based on the concentration of each gas and changing radiative efficacy, and re-run the analysis with such a dynamically updated AGTP. As for my other main comment, the framework for analysis would remain unchanged, and some or many key results may or may not change – which, again, would be a useful result in itself. All other comments are comparatively minor (though some include requests to broaden discussion or restructure some sections), as detailed below.

We thank the reviewer for this comment. (A similar one was raised by referee #2 – see response there as well.) For the well-mixed gases, adjusting radiative efficiency by background concentration is certainly possible. For $CO_2$, the dependence on emission/concentration scenarios is partly offset by/accounted for by the IRF, resulting in low scenario sensitivity (e.g., Caldeira and Kasting 1993; Aamaas et al. 2013). Due to lack of gridded scenario data, we do not include $N_2O$ in our sector/region analysis. We have however, performed an additional set of calculations which includes the dependence of methane radiative efficiency. Calculations are done using global historical and future methane (and N2O, since Etminan et al. 2016 include the overlap of methane forcing with N2O) concentration from the IIASA SSP database. For the other (not well mixed) SLCFs considered, accounting for saturation effects is more complicated, involving spatially heterogeneous cloud and chemistry interactions, and would require simulations with (or results from) complex models. Such data is not readily available and beyond the scope of the present study, and would add a significant source of uncertainty. For consistency across components, all main results are shown without the changing radiative efficiency. The discussion on methane and saturation has been included in the discussion section with a figure in the SI.

SPECIFIC COMMENTS
L83: "increase" should come after "temperature"
Corrected.
L91: "complimentary" should be "complementary" (different meaning!)
Corrected.
L96: insert "sources and" before "mitigation strategies"
Added.
L98: "inexorably" is too strong: not all SLCFs are (especially HFCs, and methane not in all regions)
We agree that this wording was not optimal. Have modified to "many SLCFs are tightly linked to"

L112: insert "co-emitted" after encompass; also, I feel it is not correct to claim that sulfate aerosols have received considerably less attention so far – certainly in the 1990s that was the dominant aerosol included in climate studies. This should be clarified a bit and some of the older literature may well be highly relevant here (e.g. focus in the US on sulfate reduction from energy systems).
Added. And we see that this sentence does not fully recognize the scientific work. We have modified the sentence to clarify that we primarily refer to assessments by e.g., UNEP, CCAC and AMAP on SLCPs: "any assessment of the potential for alleviating climate warming by SLCF reductions should encompass co-emitted species such as sulfate, not only SLCPs."

L117-121: I can't agree with that generic claim: the SRES scenarios had a wide range of evolution of methane emissions, with significant continued increases in emissions especially in the A2 scenario but also A1FI. SSPs are more nuanced but there hasn't been a material shift (unless you focus only on aerosols here – in which case, say so).
We thank the reviewer for pointing this out. We were indeed thinking primarily of aerosols and ozone precursors here. We have modified this paragraph for clarification:
"while previous scenarios for long-term evolution of aerosols and ozone precursor emissions project a general, rapid decline even in pathways with high climate forcing and GHG levels (Gidden et al., 2019; Rao et al., 2017), the most recent generation scenarios, the Shared Socioeconomic Pathways (SSPs) (O'Neill et al., 2014; Riahi et al., 2017) exhibit a much larger spatiotemporal heterogeneity in projections of these emissions. Additionally, the SSPs provide a framework for combining future climate scenarios with socioeconomic development, and hence more detailed information about

plausible future evolutions of society and natural systems. An up-to-date and detailed consideration of the emission composition is therefore timely and necessary for the design of (…)".

L160-191: As per my main comment, please expand this methodological section (possibly using SM) to demonstrate how the IRF and AGTP used in this paper compares to other IRFs. In particular, clarify whether longer-term warming contributions related to climate-carbon cycle feedbacks have been included (I argue strongly you should – tell us what the AGTP100 is for methane and HFC23). Also, add a comment here about how the AGTP adjusts over time in response to changing global GHG concentrations (again as per my main comments, I think it has to be changed dynamically to allow authors to derive conclusions about differences between SSPs/RCPs).
Please see response to general comments above. All AGTPs will also be made openly available via Figshare if the paper is accepted for publication. (Note that halocarbons are not included in this work, due to lack of available gridded and sectoral emissions data.)

L216: this section is not well structured in my view. It makes it hard to derive clear conclusions. I would suggest to improve on the structure by having one discussion about sectors, and another one about regions; also ensure you add a long-term (100 year) dimension, at present most of the discussion is for the near-term horizon.
We agree that this section could be cleaned up a it. We have made several changes to try to make it flow better. To better address the long-term dimension, we have added:
"In the long term, the net impact of AGR and WST is small, while energy is the largest individual contributor to warming due to its high $CO_2$ emissions (note that $N_2O$ is not included in the present analysis as emissions are not included in the gridded CEDS and SSP database, but would add a small contribution to the long-term impact of AGR). The second largest driver of long-term temperature change is IND, demonstrating the importance of non-$CO_2$ emissions for shaping relative weight over different time frames."

L218-223: I can see the benefits of using 10 years, but I also struggle with the claim that this is "commonly used". Especially if the authors accept my main comment, that they need to re-do their analysis with a revised IRF/AGTP, I would urge you to consider a 20-year time horizon. The reason is that (a) this is in fact commonly used (GWP20), but also (b) that 20 years puts us very close to the time when temperatures should (begin to) peak in 1.5_C scenarios – so 20 years is much more policy relevant in my view than 10 years, which is really just the near-term rate of change.
We agree that the term "commonly used" only applies to 100 years and have removed this from the sentence. We believe, however, that there are compelling arguments for and benefits of using 10 years rather than 20 as near-term (e.g., 5 year global stock take cycle, EU 2030 emission targets, 20 years being very long from the point of many investors or sectors), as the referee also notes. We do, however, provide full time series of AGTPs to allow follow-up studies to adapt to their research questions. To make this even more clear, we have added to the existing discussion of time horizons. The paragraph now reads:
"Here we select 10- and 100-year time horizons to represent near- and long-term impacts. We recognize that other choices may affect the relative importance, and even sign, of the temperature response from some of the SLCFs like aerosols and NOx, or be more relevant for certain applications. For this reason, we provide the full time series of our AGTPs (see Data Availability)."

L226/227: add a bit of nuance here: the lifetime of SLCFs varies widely, with some causing warming for many decades (methane) whereas for others the bulk of warming is in the space of a few years.
Modified to "As the impact of the SLCFs decays over years to decades upon emission (…)"

L261-277: there's a bit of confusion about whether "mitigation potential" refers to the potential to reduce the emissions of a given SLCF, or to the potential for an intervention that might affect a range of SLCFs to reduce or increase temperature in the near or long term. These are very different aspects. I would reserve the word "mitigation" for anything that focuses on the reduction of emissions of a given species, and from there discuss the implications of such actions for temperature once changes in emissions of co-emitted species are taken into account over different time frames.

Thanks for bringing this to our attention. We have made changes throughout the manuscript to be clearer and consistently use mitigation only for emission reductions, adopting the referee's suggestion.

L279: It would be really helpful if this section could clarify the scale of mitigation outcomes from SLCF mitigation compared to CO2 (and other long-lived GHG) mitigation. This would help keep the importance of SLCF mitigation in perspective, and allow the authors to use words such as "significant" with a lot more precise and justified meaning. If you only compare outcomes between SLCF mitigation approaches, but don't provide an overall scale (how much of the total mitigation in a given scenario comes from SLCFs, how much comes from CO2 and other LLGHGs), the paper could potentially be dancing on the head of a pin. You need to demonstrate how relevant this SLCF mitigation is in the bigger context (essentially a brief update from Shindell et al 2012).

Also, I feel this section needs to spell out in quite a bit more detail the assumptions behind each policy entry point and how this translates into quantified emission reductions. E.g. L285/286 says that P2 is about methane reductions, but then L305/306 seems to suggest that it can also be about CO2 reduction in the energy sector? Also more details are needed to understand the detailed emission reductions, and chemistry assumptions, for the agricultural mitigation scenarios (a lot of policies that target agricultural methane will affect agricultural N2O within farm systems). So I think the authors need to provide much more detail and quantification of how the broad policy principles in P1-P3 translate into mitigation of individual species for the different sectors. It's fine if there are subjective choices made – but we need to know what exactly those choices were to better understand to what extent the results are a function of those choices, or of the properties of the individual species that this paper helpfully aims to disentangle.

The purpose of this section is to demonstrate the applicability of our dataset for further studies of how mitigation measures and policy implementation – and, secondarily, the importance of co-emission. The policies, while loosely based on feasible measures for the sectors, are purely hypothetical and we assume that complete removal of the emissions take place. While this is to some extent described towards the end of the section, but we have now moved this clarification to the start of the section. Moreover, with this in mind, we realize that it may be confusing to use the term "policy package", when we are in fact considering packages or combinations of idealized emission reductions. The section has been rewritten for clarification, also adding more about CO2 and longer-term effects.

In addition, we have added in the final paragraph of Sect. 3.1:
"Overall, the potential for global temperature reductions inherent in the present SLCF emissions is highly inhomogeneous, and co-emitted species – including $CO_2$ – must be taken into account in any targeted climate policy for reduction of near-term warming. We emphasize that mitigation of SLCFs, while important, need to be sustained and complimentary to strong cuts in $CO_2$ for long-term reduction in global warming."

L317: add "and mitigation targets" or something like this to the section heading, as the scenarios explored are not just the SSPs but the imposition of different mitigation targets on the SSPs (i.e. they are SSPs plus climate policy). Also clarify whether the way that the mitigation of SLCFs is then implemented follows the SPA protocol developed for mitigation modelling using SSPs (Kriegler E, Edmonds J, Hallegatte S et al (2014) A new scenario framework for climate change research: the concept of shared climate policy assumptions. Climatic Change 122(3): 401-414), since this could well affect how individual SLCF emissions change for different regions.

In order to avoid making the heading to long while still capturing this point, we have modified it to: "Temperature response to SLCFs and CO2 under the SSP-RCP scenarios".

Regarding the second point, we do not explicitly model future emissions or mitigation, but use the gridded data products available via ESGF by the IAMC and extract regional emissions using a geographical mask. We realize that it is insufficiently documented and have made some addition to the methods section to clarify (adding a reference to the section in the first paragraph of Sect. 3.3): "Historical emissions are from the CEDS database, while future emissions follow the SSP-RCP scenarios. Gridded and harmonized emissions are available for nine of the SSP-RCP combinations (Gidden et al., 2019), available via ESFG from the Integrated Assessment Modeling Community (IAMC). The gridded SSP-RCP data product, including the methodology for country and sector level emission mapping, is documented by Feng et al. (2020). Regional and sectoral emission scenarios are extracted using the geographical definitions and spatial mask from HTAP2 (Janssens-Maenhout et al., 2015)."

L324: I question the utility of using SSP5-8.5 for this paper. This scenario has value but by now is clearly counterfactual as far as emissions are concerned. This would not be a critical issue, but at the same time the paper is missing a much more relevant scenario such as SSP2-2.6, or SSP5-2.6. As it stands, the only stringent mitigation scenario is for an SSP1 world, which is only one of many worlds, understanding how SLCF emissions might evolve in a different socio-economic context but also stringent mitigation would be much more valuable than to take up space for the largely academic SSP5-8.5 scenario. So, my main concern is: add a stringent mitigation scenario (RCP2.6) using a different SSP (other than SSP1), otherwise this paper is missing a really important dimension. If you then keep the 8.5 scenario or drop it is in a way secondary.

We agree that there are other scenarios in the SSP-RCP framework that could tell a different story of SLCFs in the socioeconomic context. However, to our knowledge, the gridded and harmonized emission maps are only available for the nine CMIP6 SSP-RCP combinations, which only includes SSP1 stringent scenarios. Other scenarios may have become available recently but would be beyond the timeframe and resources available for this work to add. We think this comment may partly reflect our unclear description of methods, which we have now expanded (see response to comment above). We also slightly modify Sect. 3.3:
"In the following paragraphs, we show results from four of the nine SSP-RCP scenarios used in the present analysis (SSP1-1.9, SSP2-4.5, SSP3-7.0 and SSP5-8.5). Here we choose to show the scenarios that span the range of future emission evolutions, but recognize that the realism of SSP5-8.5 is debated in the literature due to its very high emissions (e.g., Ritchie & Dowlatabadi, 2017)."

L336/337: "we note that negative CO2 emissions are not included in these calculations": I'm puzzled by this. How can you evaluate SSP1-1.9 without negative emissions? Why not? This problem would only grow if the authors follow my advice to include SSP2 or SSP5-2.6.

Thanks for pointing this out. We see that this is unclear from the description of emissions and sectors, which is insufficient and only refer to Figure 1. Our primary objective is not to evaluate SSP1-1.9 in terms of absolute temperature impact, e.g., as has been done in the recent study by Torkaska et al. 2020 (see also discussion on limitations and interpretation of our method), but to quantify and compare the sectoral and regional mitigation potential and contribution to future temperature

impact depending on whether this mitigation is achieved or not. One reason for leaving negative $CO_2$ emissions out of the analyses is that we consider it a mitigation measure, rather than a sector. From a practical point, attributing negative $CO_2$ emissions to sectors (e.g., it would in part be energy, in part forestry) is not possible from the information available in the gridded SSP-RCP emission database for CMIP6 (which we rely on here, as has also been made more clear in the methods description), as these emissions are provided as a separate category. This would make the sector comparison less transparent across components. For actually evaluating the absolute temperature response under different SSP-RCPs, we agree that the negative emissions are essential. We have therefore included them in the dataset that will be made publicly available if the paper is accepted for publication. We have also made our scope and choice clearer in the text, adding:
"We note that since our primary focus here is on quantifying the contributions to, and potential for further reduction of, near- and long-term temperature impacts, we do not include negative $CO_2$ emissions which is already a mitigation measure. Furthermore, the gridded SSP-RCP emissions only provides a separate category for negative $CO_2$ and not information for mapping the emissions to economic sectors such as energy or forestry. We do, however, include the negative $CO_2$ category in our inventory of regional scenarios for further analyses beyond our study (see Data Availability)."

Tokarska, K. B., et al. (2020). "Past warming trend constrains future warming in CMIP6 models." Science Advances 6(12): eaaz9549.

L341/342: There seems to be a rather important finding buried here: are the authors saying that globally, energy contributed less to actual temperature change than agriculture and RES? If correct this might be worth highlighting more prominently to show how including SLCFs can change relevance over different time frames. Not that this should take away from the critical importance of mitigating CO2 from ENE, but it does seem a significant element. Another study that looked at warming attributable to livestock seems to go in a similar direction (Reisinger A, Clark H (2017) How much do direct livestock emissions actually contribute to global warming? Global Change Biology DOI: 10.1111/gcb.13975).
We thank the reviewer for pointing this out and making us aware of the reference. It is indeed an interesting point that methane and other reactive gases from agriculture has had a larger temperature impact than the net effect of the energy sector. This again points to the importance of methane, as well as the role of cooling contributions from the energy sectors. We have added the reference and the following:
"The relative importance of AGR and ENE historically is yet another example of how including SLCFs can change relevance over different time frames, as also demonstrated by Reisinger & Clark (2018) for non-CO2 livestock emissions. In this example, both the warming due to CH4 from agriculture and the contributions from cooling emissions in the energy sector act to shape the relative role of the sectors over time."

L376/377: I had to read this a few times to understand the "put another way" – might be worth rephrasing or disentangling a bit
We agree that this sentence is difficult to read. Moreover, it does not add really add anything to the conclusion, and we have removed it.

L378-390: again here, as for section 3.2, I would like to see a comparison with mitigation achieved by CO2 reductions, simply to avoid readers to take away misleading conclusions that somehow SLCFs are the dominant issue for climate change – I would say they are an important but second-order issue. Useful if the paper could state and substantiate this in some way. Also for L393-395: there is "much" to be gained – how much? Compared to how much from CO2?

The relative importance of CO2 and non-CO2 contributions between the scenario can be determined from Fig.4 (for regions) and Fig.S3 (previously S1 – for sectors). To place the magnitude of temperature differences in Fig. 5 in context we have added:
"Results are shown by region and sector, for all combinations where the temperature difference is greater than ±0.01°C. For comparison, the CMIP6 mean difference between SSP3-7.0 and SSP1-2.6 (which is close to 1.9 in emissions) in projected surface temperature when accounting for all global emissions is around 0.5 °C in 2050 and 2 °C in 2100 (Tokarska et al., 2020). As seen from Fig. 4 and Fig. S3, CO2 is the key driver of this long-term temperature difference between the scenarios for most sectors and regions. However, as seen in Fig.5, there are also important SLCF contributions, most notably from the large sources of methane; agriculture, energy and waste management."

We have also made changes in several places to highlight that SLCF mitigation should only be complimentary to CO2 reductions for long-term warming reductions.

L395-422: I find this section weak on actual policy, and inconsistent: for some sectors, authors mention specific interventions, whereas for agriculture, it just says "addressing agriculture emissions" – that's not a policy or intervention. Expand this to illustrate consistently what feasible interventions are for all sectors (including a brief flag for supply vs demand side interventions).
While we acknowledge the importance of understanding how to translate the potential for climate mitigation into actual emission cuts, a detailed and comprehensive assessment of the required policy strategies is beyond the scope of the present study, as is a description of the policies that underly the SSP-RCPs, which is covered in the studies documenting respective pathways. We have made some changes to this section to streamline (e.g., adding specific examples for agriculture methane reductions) and to clarify that we here outline general features and a few examples, we have added:
"While a comprehensive assessment of policy and technological interventions required to translate this potential to actual emission cuts is beyond the scope of the present study, we outline key general features and discuss specific examples in the case of methane, referring to existing literature for additional details, in the following paragraphs. "

L424-43: this is a useful thought experiment: how much warming would be avoided simply by improving technology for SLCFs (i.e. reducing emission factors consistent with SSP1), even in the absence of any dedicated climate policy (i.e. SSP3-7.0 vs SSP3-lowNTCF).
In line with the last comment, we have also emphasized the role of technological development more in the conclusions.

L449-480: please break this discussion into chunks – lots of different issues being discussed in a single mammoth paragraph. As flagged in main comments, using nondynamic AGTP to explore SSP/RCP pathways is a real problem that the authors have to address.
We have added a sub-heading 4.1 Caveats and uncertainties and separated the following discussion into clearer paragraphs. Following the addition of a sensitivity test for methane radiative efficiency adjusted by concentration pathways (see also comment above), we have also expanded the discussion.

L464-466: agricultural non-CO2 emissions should be included in this list as they are also highly uncertain especially in developing regions (AFR, SEA, SAS).
We have added sentence to highlight that there are significant regional and sectoral differences in uncertainties in statistics and emissions:
"The level of uncertainty also differs across sectors, with emissions from nature related emissions (e.g., agriculture, landfills) more uncertain than technospheric emissions (e.g., in the fossil-fuel sector) , and regions (Amann et al., 2013; Jonas et al., 2019)."

L486: add that emission reductions of SLCFs have to be sustained to achieve longterm temperature change

We have removed the reference to long-term:
"(…) there is significant potential for additional reductions in near-term temperature change (…)"

L494-498: You could emphasise more strongly that this technological advancement brings benefits even if there is no dedicated climate policy addressing SLCFs, simply by reducing emission factors.

Yes, thank you, good point. Added.

---

## Author Comment (AC2) · 24 Jun 2020

Response to comments by anonymous referee #2 on "A continued role of Short-Lived Climate Forcers under the Shared Socioeconomic Pathways" by Lund et al.

We thank the referee for the detailed and thorough review, which has contributed to substantial improvements to our manuscript. Several steps have been taken to address the referee comments and concerns. Responses to individual comments are given below.

The manuscript emphasizes the importance of SLCF agents, especially for the short term impacts of climate scenarios, with some emphasis on methane. It is concluded that SLCFs continue to play a role in many regions. While it is important to reiterate this message, it is not so obvious what new findings are being presented. On several occasions, the results reinforce what is known, which does not justify publication.
The results for methane depend on methodological assumptions that are not transparent (e.g., emission categories) nor are they discussed in sufficient detail in the presentation of results. I found the discussion about the changing role of BC interesting, which could be highlighted more. I also recommend emphasizing regional differences more strongly. The finding that SLCFs are particularly relevant for low- and medium-income countries is relevant. In general, it would be good to deepen such analyses and bring new aspects forward more clearly.
There are some rather bold simplifications in the treatment of aerosols; e.g., it is not clear how the radiative properties of partially absorbing aerosols (with BC) are accounted for. They sensitively determine the radiative cooling efficiency. NOx is mentioned on several occasions, but its role is unclear. How is nitrate been included? It is semi-volatile and responds to changes in sulfate and ammonium. Has that been accounted for? This is particularly relevant for the comparison of scenarios.

The primary objective of this study is to provide a quantification the near- and long-term impact of individual species with a greater level of geographical and sectoral breakdown than previously existing in a unified framework, and to deliver a transparent and readily applicable data set of emission metric values for further use both in the scientific community and beyond to study the effectiveness and implications of emission changes following mitigation and policies implemented in at level of individual emission sources. We also provide the first (to our knowledge) breakdown of the SSP-RCP scenarios with this level of detail, highlighting regional evolutions that warrant further attention and work. Furthermore, following comments by referee #1 we now make a substantial methodological advancement by include the carbon-climate feedback. We have tried to make these points clearer throughout the manuscript. We have also rewritten section 3.1 to improve the flow and make the separate discussions about regions and sectors clearer, and made modifications to highlight the regional heterogeneity more clearly where possible.

In response to comments by both referees, the Methods section has been expanded to include more details about the underlying assumptions, and to guide readers outside the emission metric community. This includes e.g., specifications about AGTP for individual components and how they are treated within this concept, the choice of impulse response functions, references to the aerosol parameterizations and properties underlying the simulations of atmospheric concentrations and kernels, and emission inventories.

A relatively large temperature signal is expected from the indirect effects of aerosols on clouds, being highly non-linear especially at low pollution levels. I find the scaling by a factor of 2.1 to the impact of

sulfate questionable. I recommend investigating (and showing) how sensitive the results are toward this assumption. There could be large regional differences.

We agree that this is a simplification, and this is also discussed in the manuscript (we have modified slightly to make it even clearer). However, information about the dependence of radiative efficiency of indirect aerosol effects on emission location is to our knowledge not readily available (spatial distributions of indirect RF are of course available but would not provide the type of information we need these are typically run using all emissions as input while aerosols can travel across distances and influence clouds beyond their source region). Moreover, because we scale the regional direct radiative efficiencies, a spatial dependence is in part accounted for in the resulting AGTP for a given region, under the assumption (and that is of course not well known) that there is a similar relative influence of geographical differences in local meteorology and dynamics on both direct and indirect aerosol effect. Aerosol indirect effect are uncertain and model dependent, which poses a general challenge for climate studies across modeling tools with different level of complexity – from ESMs to emulators. The overall uncertainty in RF may well be larger than any regional difference in the efficiency. We note that we do included an analysis of the spread in our results arising from uncertainties in forcing.

l.173 mentions a lack of information. Can't you get this from the chemistry-transport model?

Generally, offline chemistry transport models do not include aerosol-cloud interactions. An estimate of the indirect aerosol forcing can be derived with subsequent radiative transfer calculations (for the first indirect effect only) but is not available to us in the form of a radiative kernel which is the approach used here. A first order estimate of the radiative forcing due to aerosol-cloud interactions has been calculated for the total global emissions by Lund et al. (2019), but similar calculations to investigate the sensitivity of the forcing to emission location (i.e., RF per unit regional emission) has not been performed and does not, to our knowledge, exist in e.g., the bulk of HTAP2 literature.

l.175: The description of the -15% for BC after l.175 is unclear (e.g., the rapid adjustment). Can you explain?

To clarify, we have modified this paragraph, which now reads:

We also account for the semi-direct effect of BC (i.e., the rapid adjustments of the atmosphere to the local heating), which has been found to partly offset the positive direct radiative forcing (Samset & Myhre, 2015). Here we use the multi-model data of the ratio between semi-direct and direct BC RF from Stjern et al. (2017) and calculate an average adjustment factor to account for the influence of rapid adjustments of -15%. This is then applied to the AGTP of BC for all regions, except South Africa where Stjern et al. (2017) found a small positive forcing from rapid adjustments.

l.190: "lower than in the literature". By how much? By 0.885/1.06? Is the effect linear?

The difference depends also on the time scales of climate response IRF, and so the difference between AGTPs using different IRFs will have a temporal dependence as well. Following this comment and a comment by referee #1 we have performed a set of sensitivity simulations for the pulse based metrics using different combinations of IRF for the climate response and CO2 to show the order of magnitude impact of our methodological choice. A separate discussion with two new figures has been added to the supplementary material.

l.200: I am doubtful about the linearization of the temperature response by multiplying the emissions with the AGTPs. There are models available to compute this properly. This is particularly relevant for aerosols and ozone (the latter not being discussed at all), and to a lesser extent for methane, which has significant indirect effects, e.g., though ozone. Has this been accounted for?

We agree that there are non-linearities in the system that are not properly represented by the AGTP approach. We also agree that there are models (i.e. coupled chemistry-climate models) that can handle this better. The problem is that these models are not suited for running experiments to quantify impacts of specific (and thus small) emissions from specific sources (by region, sector and

compound). And even the coupled models may not fully include the non-linear chemistry due to the coarse resolution of current climate models.  So, the approach by the community is to build simpler models (e.g. FaIR, Smith et al., 2018).

There are two major steps in the cause-effect chain going from emissions to temperature change. First the relation between emissions and the effective radiative forcing, and then the relation between ERF and temperature change. For the relation emission ==> ERF we have performed an additional sensitivity test that where we include the non-linear effect of methane forcing efficiency, i.e., decreasing with increasing background levels of methane (see also response to comment by referee #1). For aerosols and ozone precursors we do account for the part of the non-linear effects of emissions taking place in different regions with differences in the physical climate (e.g., temperature, radiation and precipitation) by using simulations from the HTAP experiment to calculate the em ==> conc relation for 13 global regions and then a 4-D radiative kernel to get to the global ERF. This means that our AGTPs have different values for e.g. SO2 emissions in Europe vs. South Asia because the oxidation, transport processes and removal by precipitation is different.
The part of the non-linear effect caused by the changing background levels of the pollutants in the different emissions scenarios (e.g., saturation effects in ozone chemistry or cloud responses to increasing aerosols in a higher background pollution case) is less well quantified and is not included in our analysis.

For the relation ERF ==> global temperature change we use a standard two-term impulse-response function relating global mean ERF to global mean temperature change. This has been, and still is the standard approach, in simplified climate models (and the rational for using the GWP-metric). In coupled climate models there are indications that feedbacks (and thus climate sensitivity) are state-dependent, i.e. that the sensitivity increases as the Earth warms. However, at this point, this is still not fully understood and is not well quantified at intermediate warming levels as it diagnosed from 4xCO2 experiments of CMIP6.

Smith, C. J., Forster, P. M., Allen, M., Leach, N., Millar, R. J., Passerello, G. A., and Regayre, L. A.: FAIR v1.3: a simple emissions-based impulse response and carbon cycle model, Geosci. Model Dev., 11, 2273–2297, https://doi.org/10.5194/gmd-11-2273-2018, 2018.

l.210 Mentions ozone (also l.148), but it does not appear in the rest of the manuscript. It does not show in figures 2 and 3. Why has it not been included?
As per the established emission metrics framework, temperature responses are reported in terms of the emitted species, not the subsequent forcing mechanism. The ozone precursors include the impact of ozone and methane. In addition, we include nitrate aerosols, which is only recently becoming more common. In response to this and comments above, we have added a sentence in the methods after the AGTP equation to better clarify this point to readers outside the metrics community, referring the reader to the careful documentation existing in the previous literature: "Emissions of SLCFs can have both direct and indirect radiative effects. For BC, OC and SO2 we account for the direct, semi-direct and indirect RF as described below. AGTPs for NOx, CO and VOC includes the forcing due to tropospheric ozone production and (for NOx) nitrate aerosol formation, as well as the longer-term effect on methane lifetime and methane-induced ozone loss. The AGTP for methane includes the direct forcing, as well as the effect of OH-induced changes in its lifetime and adjustments to account for indirect effects on tropospheric ozone and stratospheric water vapor. See Aamaas et al. (2013) for details and AGTP equations for individual species."

l.241: There is much debate about CH4 emissions from the fossil fuel sector. What has been assumed in the calculations, and how does it compare with recent estimates? Methane is emphasized in the conclusions, but the attribution of emissions to sectors is not transparent. It would be interesting to

deepen the discussion about the role of methane. Currently, the results are being reported but not really analyzed.

We thank the reviewer for raising this point. We use the historical, present-day and future emissions from the CEDS and SSP-RCPs inventories developed for CMIP6, and methane emissions follow the assumptions made there. From comments by both referees, we realize that the Methods discussion did not describe this very clearly and have expanded it. We also add a list of the sectors considered and their definition. While a comprehensive assessment of the influence that drive methane emissions is beyond the scope of this study, we have on several occasions added more details, following more specific comments by referee #1. The following new paragraphs have been included in the Methods section:

"Historical emissions are from the CEDS database, while future emissions follow the SSP-RCP scenarios. Gridded and harmonized emissions are available via ESFG from the Integrated Assessment Modeling Community (IAMC) for nine SSP-RCP combinations that form the core of the Coupled Model Intercomparison Project Phase 6 (CMIP6) experiments (Gidden et al., 2019): SSP1-1.9, SSP1-2.6, SSP2-4.5, SSP3-7.0, SSP3-7.0 lowNTCF, SSP4-3.4, SSP4-6.0, SSP5-3.4, and SSP5-8.5. The gridded SSP-RCP data product, including the methodology for country and sector level emission mapping, is documented by Feng et al. (2020). Regional and sectoral emission scenarios are extracted using the geographical definitions and spatial mask from HTAP2 (Janssens-Maenhout et al., 2015).

We consider the energy (ENE), agriculture (AGR), waste (WST), residential (RES), industry plus solvents (IND), transport (TRA) and shipping (SHP) sectors, as they are defined in the harmonized CEDS-SSP emission inventory (Feng et al., 2020; Hoesly et al., 2018). Due to the large spread in historical estimates and lack of emissions consistent with CEDS, we do not include emissions due to land-use/land cover. Additionally, agricultural waste burning is excluded as these are more difficult to mitigate and estimates of future $CO_2$ emissions are not available."

l.261: This is an interesting result that could be explained and emphasized more strongly.

We have expanded and added:

"These balancing characteristics do not imply that SLCF emission reductions measures should not be implemented, but that the net benefits on global temperature may be lower than expected if mitigation measures that simultaneously affect both cooling and warming SLFCs are implemented, in turn also placing added focus on the need to reduce $CO_2$ in order to mitigation warming in both the near- and long-term. Such detailed characteristics at the emission source level are needed for the design of effective mitigation strategies."

l.364-366: This is interesting and could be explained and emphasized more strongly.

We have added:

"While previous decades have seen a southeastward shift in air pollution emissions, from high income regions at northern latitudes to East and South Asia, these findings suggest that a second shift may be underway, towards low- and middle-income countries in the developing world. Further studies are needed to improve the knowledge about the resulting climate and environmental consequences, as well as how to strengthen the mitigation options, in these regions."

l.443-445: This is interesting and could be explained and emphasized more strongly.

We have expanded the explanation and the section now reads:

"Secondly, as described in Sect.2, we use an AGTP for BC that is 15% lower than in previous studies using the same methodology. This is done to account for the rapid adjustments associated with BC short-wave absorption (Stjern et al., 2017), which has been found to reduce the effective RF in a range of global climate models via changes in stability and cloud formation (Smith et al., 2018). For

our study, this factor applies to BC emissions from all sources and hence results in a reduced the net warming impact."

l.468-470: This is interesting and could be explained and emphasized more strongly.
While we agree that the recent CMIP6 results on ECS is interesting, we feel that a detailed discussion would distract from the core of the present study. We have added the reference to Zelinka et al. (2020) where the reasons for the difference in ECS estimates are discussed.